# Señorita-2M: A High-Quality Instruction-based Dataset for General Video Editing by Video Specialists

**Bojia Zi**[1,3*]     **Penghui Ruan**[2,*]     **Marco Chen**[4]     **Xianbiao Qi**[5,†]     **Shaozhe Hao**[6]
**Shihao Zhao**[6]     **Youze Huang**[7]     **Bin Liang**[1,3]     **Rong Xiao**[5]     **Kam-Fai Wong**[1,3]

[1]The Chinese University of Hong Kong     [2]The Hong Kong Polytechnic University
[3]MoE Key Laboratory of High Confidence Software Technologies, CUHK     [4]Tsinghua University
[5]IntelliFusion Inc.     [6]The University of Hong Kong     [7]UESTC

Figure 1: The visual results given by editing models trained on our Señorita-2M. *Best viewed with Acrobat Reader. Click the images to play the animation clips.*

## Abstract

Video content editing has a wide range of applications. With the advancement of diffusion-based generative models, video editing techniques have made remarkable progress, yet they still remain far from practical usability. Existing inversion-based video editing methods are time-consuming and struggle to maintain consistency in unedited regions. Although instruction-based methods have high theoretical potential, they face significant challenges in constructing high-quality training datasets - current datasets suffer from issues such as editing correctness, frame consistency, and sample diversity. To bridge these gaps, we introduce the **Señorita-2M** dataset, a large-scale, diverse, and high-quality video editing dataset. We

* Equal Contribution.
† Corresponding author. Email: qixianbiao@gmail.com

39th Conference on Neural Information Processing Systems (NeurIPS 2025) Track on Datasets and Benchmarks.

systematically categorize editing tasks into 2 classes consisting of 18 subcategories. To build this dataset, we design four new task specialists and employ or modify 14 existing task experts to generate data samples for each subclass. In addition, we design a filtering pipeline at both the visual content and instruction levels to further enhance data quality. This approach ensures the reliability of constructed data. Finally, the **Señorita-2M** dataset comprises 2 million high-fidelity samples with diverse resolutions and frame counts. We trained multiple models using different base video models, i.e., Wan2.1 and CogVideoX-5B, on Señorita-2M, and the results demonstrate that the models exhibit superior visual quality, robust frame-to-frame consistency, and strong instruction following capability. More videos are available at: `https://senorita-2m-dataset.github.io`.

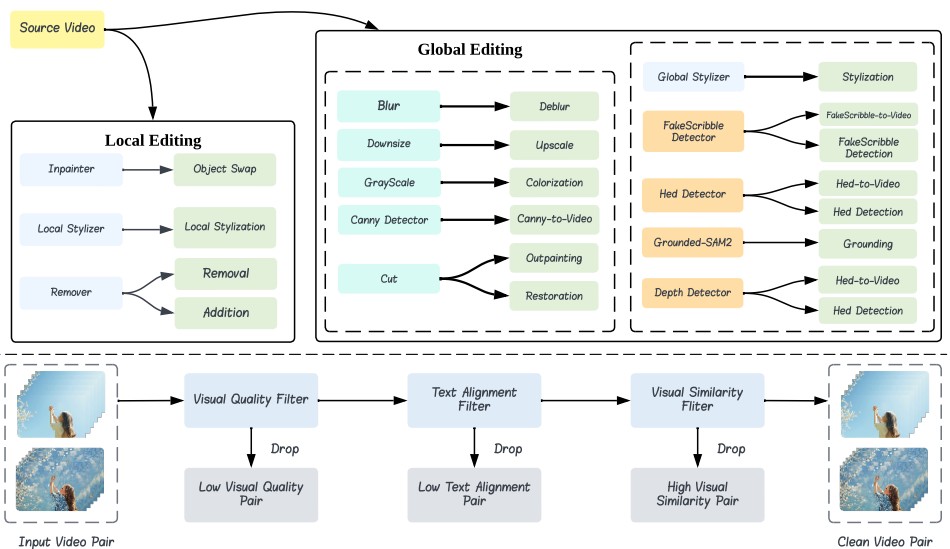

Figure 2: Top: Editing tasks in Señorita-2M dataset. Bottom: The filtering pipeline of Señorita-2M. Further details are provided in the Appendix. In the top subfigure, the blue rectangle represents the experts trained by us, the cyan rectangle represents the transforming operation, while the yellow rectangle is the existing experts.

# 1   Introduction

Editing is a critical task in the field of computer vision [12, 6, 50, 42, 20, 54, 46, 22, 53, 56, 21]. In recent years, with the rapid development of diffusion-based generative models, editing techniques have also advanced rapidly. While image editing has been extensively researched and achieved remarkable results, video editing—a more recently explored area—still requires further development to attain satisfactory outcomes. Existing video editing techniques are predominantly inversion-based [12, 36], which employ inversion methods to reconstruct noise and perform editing during the denoising process. These approaches suffer from two significant drawbacks: they are highly time-consuming, and more importantly, it is challenging to ensure the consistency of unedited regions with the original video.

Another category of instruction-based methods has therefore garnered increasing attention [6, 47, 20, 42, 54, 46, 5, 11]. Unlike inversion-based approaches, which are training-free, these methods rely on datasets consisting of [original video, edited video, instruction] triplets. They train an end-to-end model using such datasets, where the model takes unedited videos and corresponding instructions as inputs and outputs edited videos. By leveraging deep models' powerful learning capabilities through training on data, rather than being training-free like inversion-based methods, instruction-based approaches have high theoretical ceilings. Thus, a critical challenge for instruction-based methods lies in constructing high-quality training datasets.

Several works have attempted to construct such datasets. InsV2V [8] generates source and target videos along with corresponding instructions by using the same noise and different prompts. However, since both the original and edited videos are model-generated, their quality is low. Additionally, the dataset suffers from low resolution and a limited number of training samples. InsViE [47] starts with real videos, edits the first frame using image editing techniques, generates edited videos from the modified first frame via an Image-to-video model, and finally applies optical flow-based consistency filtering. This approach struggles to ensure consistency between the target and source videos, often leading to drastic changes in the edited video compared to the original. VIVID [18] trains an inpainting model to perform editing operations (e.g., object addition/removal) on real videos, but the dataset covers a narrow range of editing types, lacks diversity, and is not open-source.

To address these challenges, we introduce the **Señorita-2M** dataset, which is more diverse, reliable, and faithful. We categorize editing tasks into two broad classes: local editing and global editing, further systematically subdividing them into 18 subcategories. For these 18 subcategories, we employ different specialists to generate [original video, edited video, instruction] triplets. Some specialists are specially trained by us, while others are implemented using subtle techniques based on existing mature algorithms, with detailed elaboration provided in Section 3. These specialists ensure the correctness and reliability of samples within each edit subcategory. To further enhance triplet quality, we design a filtering pipeline to effectively remove failed triplets: (1) we manually annotate and train an editing quality scoring model; (2) we further filter the dataset based on the alignment between editing effects and instructions, as well as the degree of editing modifications.

Our main contributions can be summarized into three folders:

- We introduce the high-quality **Señorita-2M** dataset for training instruction-based editing models, which is diverse, reliable, and faithful. It contains 2 million high-fidelity samples of source and target videos with corresponding instructions, featuring diverse resolutions, frame counts, and has been open-sourced.

- We systematically categorize editing tasks into 2 broad classes and 18 subcategories to ensure diversity. For each subcategory, we design specialized specialists to generate samples individually, guaranteeing data reliability. Additionally, a filtering pipeline is developed to further enhance reliability.

- Our dataset enables training of extremely high-quality video editors across different base models. The resulting model exhibits superior visual quality, robust frame-to-frame consistency, and strong alignment with text instructions.

Table 1: Comparison between Señorita-2M, InsV2V, InsViE and VIVID.

| Datasets | Sources | Editing Types | Frames | Resolution | Edited Pairs | OpenSource |
|---|---|---|---|---|---|---|
| InsV2V [8] | Synthesis | Free-Form | 16 | $256 \times 256$ | 0.06M | ✔ |
| InsViE [47] | Real Videos | Free-Form | 25 | $576 \times 1024$ | 1M | ✔ |
| VIVID [18] | Real Videos | 3 Categories | 30 | $720 \times 1280$ | 1.5M | ✗ |
| Señorita-2M | Real Videos | 18 Categories | 33 - 64 | $336 \times 592$ - $1120 \times 1984$ | 2M | ✔ |

## 2 Related Works

### 2.1 Image Editing.

Recent image editing methods have emerged [12, 36, 44, 19, 34, 45], which can be broadly categorized into two types: *inversion-based* and *instruction-based* approaches. Inversion-based methods work by converting image latents into noise and regenerating the image with a new prompt. Although these methods are training-free, they are often time-consuming and struggle to preserve unedited regions. In contrast, instruction-based methods train editors that take explicit instructions to edit images, enabling faster inference and more reliable results. Early instruction-based models, such as InstructPix2Pix [6], HIVE [52], and MagicBrush [50], were trained on medium-scale datasets. More recently, methods like HQ-Edit [20], Omni-Edit [46], and Emu-Edit [42] leverage large-scale datasets to produce high-quality and visually pleasing edits.

## 2.2 Inversion-based Video Editing.

Most existing video editing approaches are based on inversion techniques [40, 29]. Methods such as Pix2Video [7] and TokenFlow [13] perform frame-wise inversion of video frames into noise and aim to preserve temporal consistency through attention mechanisms or keyframe-guided editing strategies. Flatten [9] combines DDIM inversion with optical flow to facilitate video editing. AnyV2V [24], in contrast, injects guidance features from the edited first frame to influence the generation of subsequent frames. Despite considerable efforts, current inversion-based methods still face several challenges: (1) unintended modifications in regions that should remain unaltered; (2) difficulty in maintaining temporal consistency across frames, especially over long time spans.

## 2.3 Instruction-based Video Editing.

In contrast, only a few works adopt instruction-based video editing [43, 49, 30]. VIVID [18] focuses on local edits and introduces a dataset with 1.5 million video samples. However, they include only three local editing tasks and cannot be applied to a broader range of editing scenarios. Differently, InsV2V [8] and InsViE [47] address on wide range of editing tasks. InsV2V utilizes synthetic videos generated by VideoP2P[31], while InsViE applies image editing on the first frame and uses SVD [4] to propagate the changes across the video. However, since both datasets are mainly curated by outdated generation models instead of editing experts, the motion consistency, visual quality and text-video alignment couldn't be guaranteed.

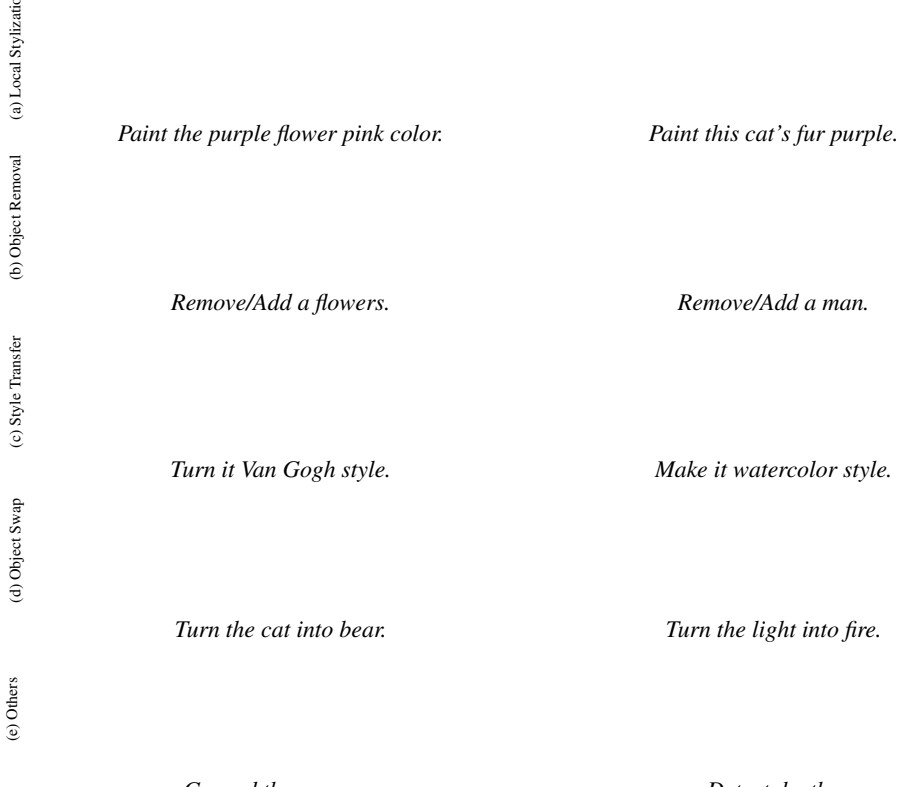

Figure 3: Visualization of our Señorita-2M. *Best viewed with Acrobat Reader. Click the images to play the animation clips.*

## 3 Methodology

In this section, we provide a detailed introduction to the proposed Señorita-2M dataset. Specifically, we first describe the preparation of data sources, then elaborate on how we construct the training

data for local and global editing samples, i.e., the 18 subcategories, and finally present our designed filtering pipeline.

## 3.1  Data Preparation

We crawled videos from Pexels [38] by authenticated APIs, which is a video-sharing website with high-resolution and high quality videos, without copyright dispute. Our dataset comprises approximately 390,000 real videos. Following video collection, we employed BLIP-2 [26] for video annotation. For local editing tasks requiring object detection to locate editing regions, we utilized CogVLM2 [17] and Grounded-SAM2 [32, 41] to generate precise masks and corresponding captions for the target regions. Subsequently, we will utilize this dataset to construct [original video, edited video, instruction] triplets for our research framework.

## 3.2  Editing Tasks

We categorize editing tasks into two broad classes: local editing and global editing. The subcategories under each class, along with their corresponding percentages, are presented in Table 2. Below, we provide a detailed description of these video edit tasks.

Table 2: The subcategories of local editing and global editing, along with the percentage of sample counts in the total dataset. *-2V means *-to-Video, Inpainting is the Video Inpainting.

| Task | Style Transfer | Object Addition | Object Removal | Local Stylization | Outpainting | Colorization |
|---|---|---|---|---|---|---|
| Types | Global | Local | Local | Local | Local | Global |
| % | 3.1 | 26.6 | 20.7 | 17.5 | 2.0 | 0.7 |
| Task | FakeScribble | Upscale | Object Swap | Hed Detection | Hed-2V | Depth Detection |
| Types | Global | Global | Local | Global | Global | Global |
| % | 1.1 | 0.3 | 8.9 | 1.1 | 1.1 | 1.1 |
| Task | Depth-2V | Canny-2V | Deblur | FakeScribble-2V | Inpainting | Grounding |
| Types | Global | Global | Global | Global | Local | Global |
| % | 1.1 | 1.1 | 0.3 | 1.1 | 1.2 | 11.0 |

### 3.2.1  Local Editing

Local editing refers to precise modifications within specific video regions while preserving the surrounding content. It includes 6 key tasks: object swap, object addition, object removal, local stylization, video inpainting, and video outpainting. These techniques allow for swapping or adding objects, removing unwanted elements, altering appearances, repairing damaged areas, and extending videos beyond their original boundaries. Note that the object swap, object addition/removal and local stylization are crafted by our trained experts, the training details can be found in Appendix.

**Object Swap.** Object swap refers to replace a specific object in a video with another object. Our expert involves an image inpainter [3] to edit first frame and a video inpainter trained by us to inpaint the rest frames guided by the first edited frame. In practice, we treat the edited videos as the source and the original videos as the target to preserve the highest possible video quality for the final output. Once the object is swapped within the specified region, we prompt the LLM [10] to generate coherent editing instructions, referencing both the original and the replacement object names. This process results in a set of original videos, edited videos, and associated instructions.

**Object Removal and Addition.** Object removal and addition refer to the operation of removing an object from the original video or adding a new object to it. Conventional inpainting models tend to generate new objects in the inpainting region, whereas a remover requires the precise elimination of objects from the specified area. To address this, we trained a specialized remover model designed exclusively for object removal, ensuring that no new content is synthesized within the masked areas. When inference, we perform object removal using the trained remover with given object regions. Once the removal is completed, we set the original video as the source video and the edited video as

the target video. The removal instruction generated by a LLM is paired with these videos to form a removal triplet. In contrast, for object addition, we reverse the roles: the edited video becomes the source, and the original video serves as the target.

**Local Style Transfer.** Local style transfer involves modifying the style of specific regions within a video. To accomplish this, we trained a local stylizer—an inpainting model built upon the ControlNet, where the inpainting model aims to modify the local region and preserve the background unchanged, the ControlNet module is responsible for using the control conditions(e.g., Canny Edges) to style transfer. When constructing video data for local stylization, we first prompt a LLM to generate new descriptions with style information based on the given objects. These updated descriptions, along with the masks and control conditions, are then fed into our local stylizer to repaint the object with different appearance. Finally, the descriptions are converted into instructions for local stylization by using LLM, completing the process.

**Inpainting and Outpainting.** Inpainting refers to repairing missing pixel regions in the original video, while outpainting refers to extending the video's edge pixel regions to expand the frame boundary. Both tasks are straightforward to construct training data for and involve no model-dependent operations. For Video Inpainting, we first remove a moving region from the original video by setting its pixels to black (pixel value 0). The video with the missing region serves as the source video, and the original video is used as the target video to supervise model training. For Outpainting, the process is similar, except that the region removed is the edge pixel area of the video, enabling frame expansion. Instructions for both tasks are generated by prepending "`restore this video.`" and "`outpaint this video.`" to the original caption.

### 3.2.2 Global Editing

Global editing refers to editing tasks that modify the entire video. It encompasses 12 subcategories, including style transfer, object grounding, and a series of controlled video translation tasks: deblur, upscaling, colorization, depth detection, HED detection, FakeScribble detection, depth-to-video, HED-to-video, FakeScribble-to-video, and canny-to-video. Note that the style transfer is performed by our trained global stylizer, while the remaining tasks are either edited using existing specialists or generated directly. For the training details of global stylizer, you can find in the Appendix.

**Style Transfer.** Style transfer refers to the task of applying an artistic style on a given video. Therefore, we trained a global stylizer, a first-frame guided video ControlNet model. During inference, to construct style transfer triplets, we first craft the style prompts by combining style descriptions provided by Midjourney [35] with video captions generated by BLIP-2. Then, we apply image ControlNet [51] to stylize the first frame based on a given style prompt. Subsequently, our first-frame-guided video ControlNet propagates the stylized first frame to the remaining frames of the video. Finally, we use a LLM to convert the style prompts into natural language instructions.

**Object Grounding.** Object Grounding can be viewed as a video-to-video process where the input is the original video, and the output is a video containing grounding information for specific objects, i.e., directly visualizing grounding results in video form. Thus, this can be also recognized as a video editing task. In this setup, video editors can understand and localize relevant regions simply through instructions, where areas unrelated to the prompt are masked in black, while prompt-relevant instances are highlighted in distinct colors. To construct video pairs, we directly utilize Grounded-SAM2. Initial instructions are generated by prepending words like "Detect" or "Ground" to the object name, after which an LLM refines these instructions to enhance accuracy and clarity.

**Controlled Video Translation.** Controlled video translation represents a specialized video-to-video task. As noted in Object Grounding, it can be broadly classified as a video editing task. This section describes 10 distinct tasks designed for video translation under specific control conditions. In deblur and upscaling, the source videos are degraded (either blurred or low-resolution), while the targets are their high-quality originals. For colorization, grayscale videos serve as the source, with RGB-channel videos as the target. Tasks such as depth detection, HED detection, and FakeScribble detection leverage classical computer vision techniques—such as depth estimation, hed detection, and scribble extraction—to transform the original video into a target representation (e.g., depth maps, hed maps, or scribble annotations). Similarly, tasks like Depth-to-Video, HED-to-Video, FakeScribble-to-Video, and Canny-to-Video reuse these source-target pairs for video generation. All task instructions are further refined using a LLM for enhancement.

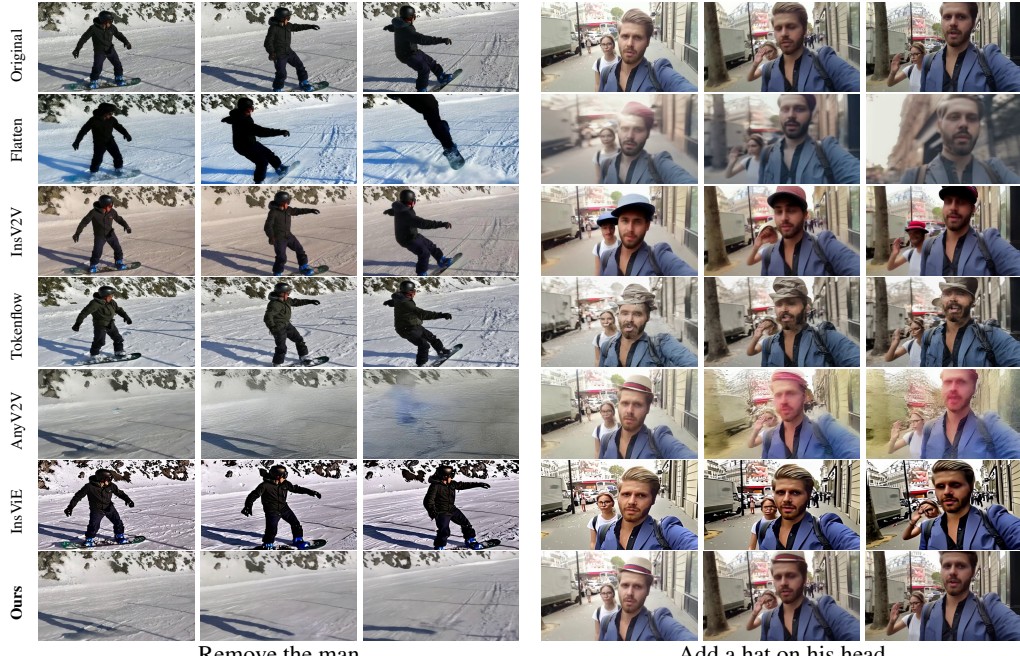

Figure 4: Editing results compared between different editing methods.

## 3.3 Data Selection

We proposed a filtering pipeline to select proper edited videos and further enhance the reliability of triplets in the dataset. First, we trained a scoring model for quality filtering to evaluate and recognize successful edits. Next, we conducted an instruction alignment assessment to remove edits that do not match the specified instructions. Finally, we excluded data where the pre- and post-editing videos exhibit minimal or no changes.

**Quality Filtering**. We built classifiers to filter out failed editing cases. To this end, we manually selected 3K successful edits (i.e., high-quality triplets) and 2K failed edits (i.e., low-quality triplets with inconsistent motion or artifacts), then trained classifiers on this manually curated dataset. Specifically, we use a frozen CLIP vision encoder to extract features from 17 frames per video, concatenate the embeddings, and feed them into a MLP as the classifier. To enhance robustness, three such classifiers are ensembled. During filtering, task-specific thresholds are applied to ensure effective discrimination.

**Removing Poor Text-alignment Videos**. We observed that some generated videos did not align well with their corresponding instructions. To address this issue, we utilized CLIP to compare target samples with their respective instructions. Similarly, in practice, we applied task-specific thresholds for filtering. We started with a low threshold and gradually increased it, evaluating the filtering results at each step. Once no significant improvement in alignment was observed, we selected the current threshold for use.

**Removing Subtle Video Pairs**. Some video pairs show subtle edits or regenerate content similar to the original. These could cause overfitting during training. To filter these out, we used CLIP's vision encoder to extract features and compare the original and edited videos, excluding pairs with a similarity score above a set threshold.

## 4 Experiments

In this section, we first introduce the training settings and inference details of the editor trained on our **Señorita-2M** dataset. We then compare our editor with baselines with quantitative and qualitative results. Additionally, we conduct ablation studies, validate our dataset across different base video foundation models and explore various editing architectures.

**Training Settings and Inference Details**. We use CogVideoX-5B-I2V [48] as the base model and integrate it with ControlNet. In this framework, the first-frame-guided main branch leverages the edited first frame to steer the video editing process, while the control branch injects source information into the main branch to ensure consistency and guidance. The batch size of the editing model's training is 32, the learning rate is 1e-5, and weight decay is 1e-4. We train model for 2 epoch. We sample 33 frames of the videos to train with a resolution of 336×592 in first stage. Different from the first stage, we use higher resolution of 448×768 and batch size of 16 in stage two, fintuning with 1 epoch to help model edit high resolution. We inference a video on 4090 GPU with 50 steps needs 5 minutes by editor based on CogVideoX-5B-I2V, 33 frames and resolution of 448×768.

**Evaluation Metrics**. For evaluation, we utilize the DAVIS dataset [39] with randomly generated editing prompts. We evaluate the stability of the edited videos using Ewarp and Temporal Consistency, while the CLIPScore is used to assess the text-video alignment. To evaluate Ewarp, we use a resolution of $336 \times 592$. Temporal Consistency is measured by extracting features from each frame and computing the similarity between adjacent frames. We conducted a user study in which participants were asked to select their preferred video in a total of 30 comparison questions.

## 4.1 Experimental Results

We compared the editor trained on our dataset with previous editing methods to demonstrate the effectiveness of our dataset. Additionally, we conducted an ablation study to show that our editing dataset can effectively produce a powerful editor. Besides, we proved the generalization of our dataset across different base models. Furthermore, we conducted experiments by training 6 models with different architectures to understand the impact of architectures on editing.

Table 3: Comparison with previous methods. **Temp-Cons** is the abbreviated form of Temporal Consistency. The best results are **blodfaced**.

| Methods | Ewarp($10^{-3}$) ($\downarrow$) | CLIPScore ($\uparrow$) | Temp-Cons ($\uparrow$) | User Preference ($\uparrow$) |
|---|---|---|---|---|
| Tokenflow [13] | 16.31 | 0.2637 | 0.9752 | 6.30% |
| Flatten [9] | 16.31 | 0.2461 | 0.9690 | 4.81% |
| AnyV2V [24] | 20.48 | 0.2723 | 0.9709 | 11.11% |
| InsV2V [8] | 16.50 | 0.1675 | 0.9727 | 16.30% |
| InsViE [47] | 33.89 | 0.1826 | 0.9695 | 5.92% |
| **Señorita-Editor** | **9.42** | **0.2895** | **0.9775** | **55.56%** |

**Quantitative Comparison.** Based on the provided experimental results in Table 3, our method outperforms all others across several metrics. Specifically, in terms of Ewarp, our method achieves the lowest value at 9.42, significantly outperforming Tokenflow, Flatten, InsV2V, AnyV2V and InsViE, which have higher Ewarp values. In terms of CLIPScore, our method also leads with a score of 0.2895, surpassing the highest score from AnyV2V, which is 0.2723. Furthermore, our method excels in Temporal Consistency with a score of 0.9775, higher than the other approaches.

To further demonstrate the effectiveness of our dataset, we compare our results with two instruction-based methods- InsV2V and InsViE, which require video triplets to train editors. In terms of Ewarp, InsV2V achieves a value of 16.50, significantly worse than our 9.42. In CLIPScore, InsViE scores 0.1826, which is notably lower than our 0.2895. Additionally, in Temporal Consistency, InsV2V scores 0.9727, whereas our method leads with 0.9775. These comparisons highlight that our approach consistently outperforms InsV2V across all metrics, demonstrating the superior quality and effectiveness of our dataset in achieving better editing performance.

**Qualitative Comparison.** We conducted a user study to determine which video users preferred. The sequence of videos in the questionnaire was randomly shuffled to ensure fairness. As shown in Table 3, our method significantly outperforms all previous approaches, achieving an impressive user preference score of 55.56%, which is notably higher than the next best score of 16.30%. This highlights the superior appeal and relevance of our method in meeting user needs. Moreover, the results showcase the effectiveness of our datasets, solidifying their superiority over existing alternatives. The Figure 4 shows the comparison between our editors with all baselines. It can be obviously found that our method have better visual quality and consistency, while doesn't show artifacts.

Table 4: The results of the ablation study are presented. All three models are fine-tuned based on CogVideoX-5B. The best results are **blodfaced**.

| Methods | Dataset | Ewarp $(10^{-3})$ ($\downarrow$) | CLIPScore ($\uparrow$) | Temporal Consistency ($\uparrow$) |
|---------|---------|----------------------------------|------------------------|-----------------------------------|
| Ablation-1 | InsV2V [8] | 8.51 | 0.2366 | 0.9712 |
| Ablation-2 | InsViE [47] | 15.72 | 0.2117 | 0.9607 |
| Ablation-3 | Señorita-2M | **8.44** | **0.2596** | **0.9783** |

Table 5: The generalization of Señorita-2M on different base models. The first model is the diffusion model, while the rest models are flow matching models. The best results are **boldfaced**.

| Base Model | Modeling | Ewarp $(10^{-3})$ ($\downarrow$) | CLIPScore ($\uparrow$) | Temp-Cons ($\uparrow$) |
|------------|----------|----------------------------------|------------------------|------------------------|
| CogVideoX-5B-I2V | Diffusion | **9.42** | 0.2895 | **0.9785** |
| Wan2.1-1.3B | Flow Match | 12.80 | 0.2928 | 0.9781 |
| Wan2.1-14B-I2V-480P | Flow Match | 10.70 | **0.2953** | 0.9732 |

## 4.2 Understanding of Our Dataset

**Ablation Study**. The ablation study in Table 4 demonstrates that utilizing samples from the Señorita-2M dataset enhances the model's performance. Since InsV2V only has 60K video triplets, we thus random extracted 60K video triplets the rest two dataset to keep fair comparison. We trained these dataset with 8 epochs on CogVideoX-5B. Compared to experiments using videos from InsV2V and InsViE dataset, the experiment with Señorita-2M, yields superior results. Specifically, the CLIPScore improves from 0.2117, and Temporal Consistency increases from 0.9607 to 0.9783. Our dataset also achieves best Ewarp value with 8.44.

**Generalization Across Different Base Models**. We also evaluated the dataset generalization across different base models, as shown in Table 5. The models we chosen are CogVideoX-5B-I2V, Wan2.1-1.3B and Wan2.1-14B-I2V-480P, where the first model is the diffusion model, the rest two models are flow matching [28] models. It can be found that, the Wan2.1-14B-I2V-480P achieves better performance in the Ewarp and CLIPScore, while slightly lower than Wan2.1-1.3B on Temporal Consistency. For the base model CogVideoX-5B-I2V, it achieves best Ewarp and Temporal Consistency, but lower CLIPScore. Overall, all three base models produce considerable results, the dataset can be used with different architectures and base models.

Table 6: Exploration of different editing architectures. *Ins-Edit* refers to the InstructPix2Pix architecture, *Control-Edit* denotes the ControlNet architecture for video editing. * indicates the use of the Omni-Edit dataset for enhancement. *FF-* are first-frame guided editing models. The best results are **boldfaced**.

| Methods | Ewarp $(10^{-3})$ ($\downarrow$) | CLIPScore ($\uparrow$) | Temp-Cons ($\uparrow$) | User Preference ($\uparrow$) |
|---------|----------------------------------|------------------------|------------------------|------------------------------|
| Ins-Edit | 13.18 | 0.2648 | 0.9797 | 3.87% |
| Control-Edit | 12.81 | 0.2882 | 0.9769 | 14.40% |
| Ins-Edit* | 13.83 | 0.2789 | 0.9784 | 8.86% |
| Control-Edit* | 10.46 | 0.2866 | **0.9802** | 23.26% |
| FF-Ins-Edit | **8.44** | 0.2861 | 0.9783 | 12.46% |
| FF-Control-Edit | 9.42 | **0.2895** | 0.9775 | **37.12%** |

## 4.3 Exploration of Effective Editing Architectures

We explore different editing architectures by training 6 instruction-based models with various architectures. We draw on two widely used image editing architectures, InstructPix2Pix [6] and ControlNet [51]. Specifically, InstructPix2Pix concatenates conditions and input latents, outputting predicted noise, while ControlNet uses a control branch for editing conditions and a main branch for input latents. We also investigate strategies with and without first frame guidance. Additionally, we explore the editing results with or without the enhancement of image editing dataset [46]. The

results are shown in Table 6. Editors based on controlnet with image enhancement achieves best temporal consistency, while the editors with first-frame guidance have best Ewarp and CLIPScore at 8.44 and 0.2895, demonstrating that the first-frame guidance has superior performance and the image enhancement can build stronger editor. Besides, more users choose the videos edited by FF-Control-Edit and Control-Edit*, which means the ControlNet is best suited than other candidate architectures.

Figure 5: More visual results given by editing models trained on our Señorita-2M. *Best viewed with Acrobat Reader. Click the images to play the animation clips.*

## 5   Conclusion

In this paper, we trained a set of advanced video editing models and integrated them with various computer vision experts to create a large scale, diverse and reliable instruction-based video editing dataset. This dataset includes 18 distinct video editing tasks, comprising approximately 2 million video pairs in various resolutions and frame lengths. To ensure best reliability, we applied multiple filtering algorithms to check and filter the failure cases. Additionally, we employed a large language model to transform editing information into precise editing instructions. Experimental results demonstrate that high-quality video editors can be effectively built using our dataset across various base models, producing frame-consistent, visually pleasing and instruction-following edited videos.

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

# A   The Design of Video Editing Experts

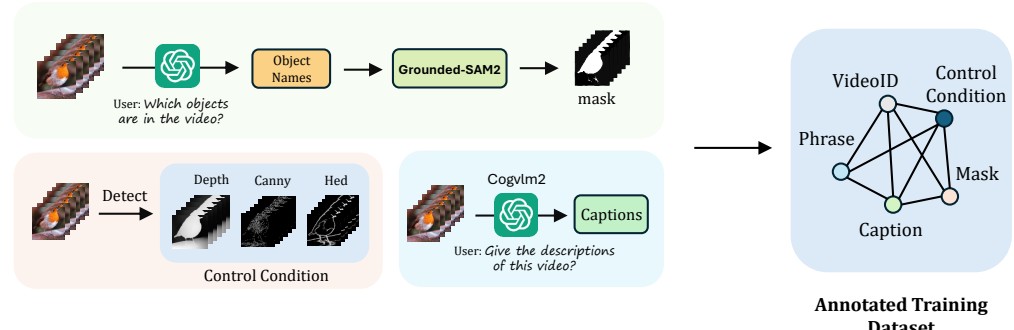

Figure 6: The construction pipeline for the annotated dataset used in expert training. Each annotated sample includes: [video, mask, object name, Canny map, HED map, depth map, caption]. We use CogVLM2 [17] and Grounded-SAM2 [32, 41] to obtain our annotations. This curated dataset serves as the training dataset for our experts.

## A.1   The Construction of Expert Training Dataset

As shown in Figure 6, we built a well-annotated dataset based on watermark-free **WebVid-10M** [1] to train our expert models. We use the CogVLM2 [17] to recognize objects in the video and feed these object names into Grounded-SAM2 [32, 41]. This process generates phrase names and corresponding object mask sequences within the video. Moreover, We use Canny, HED and depth detector to get control conditions. We also use CogVLM2 [17] to get detailed caption of each video. All data annotation processes are conducted on 4090 GPUs.

## A.2   The Construction of Global Stylizer

Table 7: Quantitative Comparison on Global Stylization. Temp-Cons is the temporal consistency. The best results are **boldfaced**.

| Methods | Ewarp$(10^{-3})(\downarrow)$ | CLIPScore ($\uparrow$) | Temp-Cons ($\uparrow$) |
|---|---|---|---|
| Tokenflow [13] | 19.99 | 0.3125 | 0.9752 |
| Flatten [9] | 11.18 | 0.3127 | 0.9759 |
| InsV2V [8] | 9.61 | 0.2864 | 0.9736 |
| AnyV2V [24] | 34.94 | 0.2928 | 0.9687 |
| VACE [21] | 11.18 | 0.2913 | **0.9797** |
| Our Expert | **9.02** | **0.3145** | 0.9781 |

We find that two powerful video generation models, e.g., CogVideoX [48] and HunyuanVideo [23], lack sufficient ability to generate videos that accurately follow style information. This limitation prevents us from applying techniques such as ControlNet to repaint a video effectively. To address this, we shift our focus to image-based ControlNet to leverage the strong stylization capabilities of these models, enhancing the stylization of video generation. Specifically, we first apply an image ControlNet to process the first frame, then use a video ControlNet to propagate the style across the remaining frames. Since video generation models inherently maintain temporal consistency between frames, the style applied to the first frame can be effectively transferred to the rest of the video.

**Architectures**. We integrate DiT-ControlNet [37, 51] and CogVideoX-5B-I2V [48] to propagate the first-frame style consistently throughout the entire video.

As shown in Figure 8, the main branch first concatenates [nosiy latents, padded latents] and feeds the result into a patch embedder. In this context, the *padded latents* retain information from the first frame, while the *latents* refer to noisy inputs with added noise. The control branch consists

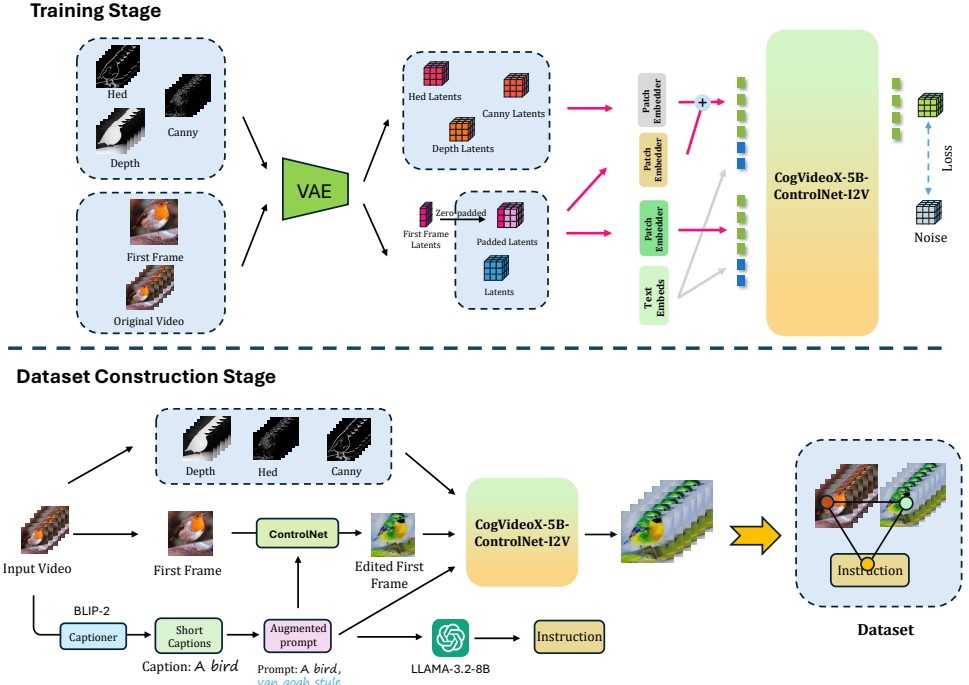

Figure 7: The training pipeline for our global stylizer comprises a main branch and a control branch. The main branch receives the concatenation of a zero-padded latent and a noisy latent, resulting in a tensor with shape $2c \times f \times h \times w$. In the control branch, one patch embedder processes the concatenated latent, including [HED, Canny, depth latents], with shape of $3c \times f \times h \times w$. A second patch embedder in the control branch is fed exactly the same input as the main branch. Our training target is to reconstruct the latent code of the original video. In the training stage, the first frame is taken from the original training video. In the inference stage, the first frame is a stylized image generated by ControlNet-SD-1.5 [51].

of two patch embedders: the first has the same architecture as the main branch's embedder but uses independent parameters, and the second is dedicated to processing control information. The outputs from both embedders are summed and then passed into the DiT block.

The control condition injection process is structured such that the main branch comprises $M$ layers and the control branch has $N$ layers, where $M = T \times N$ and $T = 6$. Control features are injected into the main branch using zero convolution. Specifically, the $K$-th layer of the main branch receives the hidden state from the $(K \bmod N)$-th layer of the control branch.

**Training Details.** We initialize the main branch with CogVideoX-5B-I2V weights and clone its first six DiT blocks to form the control branch, zero-initializing only the control patch embedder. The global stylizer is trained for one epoch on an expert dataset (batch = 8, learning rate = $1 \times 10^{-5}$, weight decay = $1 \times 10^{-4}$). To contain computational cost while preserving generalization, normalization and FFN layers in the backbone remain frozen, whereas the control branch updates only its first DiT block, patch embedder, and attention layer. Training proceeds in two phases: Phase-1 trains on $256 \times 448$ videos with 33 frames and use 10% null prompt to enable classifier-free guidance; Phase-2 fine-tunes the model from Phase-1 with higher resolution for high quality style transfer.

**Inference Details.** During inference, we append the required style prompt to the end of the video description, creating a new combined prompt. The first frame is generated by ControlNet-SD1.5, which is then fed into the model along with the prompt and control condition. We use the classifier-free guidance of 4. The model processes a video within 2 minutes on a 4090 GPU, at a resolution of $336 \times 592$, producing 33 frames.

**Evaluation.** As shown in As shown in Table 7, our expert model outperforms all baseline models, achieving the lowest Ewarp score of 9.02 and the highest CLIPScore of 0.3145. While InsV2V

*Make it Van Gogh style.*

*Make it Ghibli style.*                    *Make this video Ghibli Style.*

*Make it Van Gogh style.*                  *Make it Van Gogh style.*

*Make it Watercolor style.*                *Make it Watercolor style.*

*Make it Cyber Punk style.*                *Make it Cyber Punk style.*

Figure 8: The visual results of our global stylization. The video on the left depicts the original video, while the video on the right displays the edited videos. *Best viewed with Acrobat Reader. Click the images to play the animation clips.*

demonstrates strong performance in terms of Ewarp, it lags in text alignment. VACE exhibits a slightly higher temporal consistency value of 0.9797, but its low CLIPScore renders it unsuitable for style transfer. These results highlight the superior balance of our expert model across visual quality, text alignment, and temporal smoothness.

## A.3 The Construction of Local Stylizer

Table 8: Quantitative Comparison on Local Stylization. The best results are **boldfaced**. Temp-Cons is the temporal consistency.

| Methods | Ewarp($10^{-3}$) ($\downarrow$) | CLIPScore ($\uparrow$) | Temp-Cons ($\uparrow$) | PSNR ($\uparrow$) | SSIM ($\uparrow$) |
|---|---|---|---|---|---|
| Tokenflow [13] | 16.60 | 0.2876 | 0.9810 | 18.79 | 0.8555 |
| Flatten [9] | 17.18 | 0.2923 | 0.9751 | 18.64 | 0.8605 |
| InsV2V [8] | 7.40 | 0.2830 | 0.9783 | 20.81 | 0.9091 |
| AnyV2V [24] | 15.77 | 0.2920 | 0.9759 | 19.60 | 0.8884 |
| Our Expert | **6.50** | **0.2944** | **0.9828** | **28.29** | **0.9843** |

Inspired by SparseControl [15], COCOCO [56], and AVID [53], we trained a local stylizer by combining both inpainting and ControlNet, enabling appearance modification, stylization, and texture manipulation in specific regions of videos, while keeping the original background unchanged.

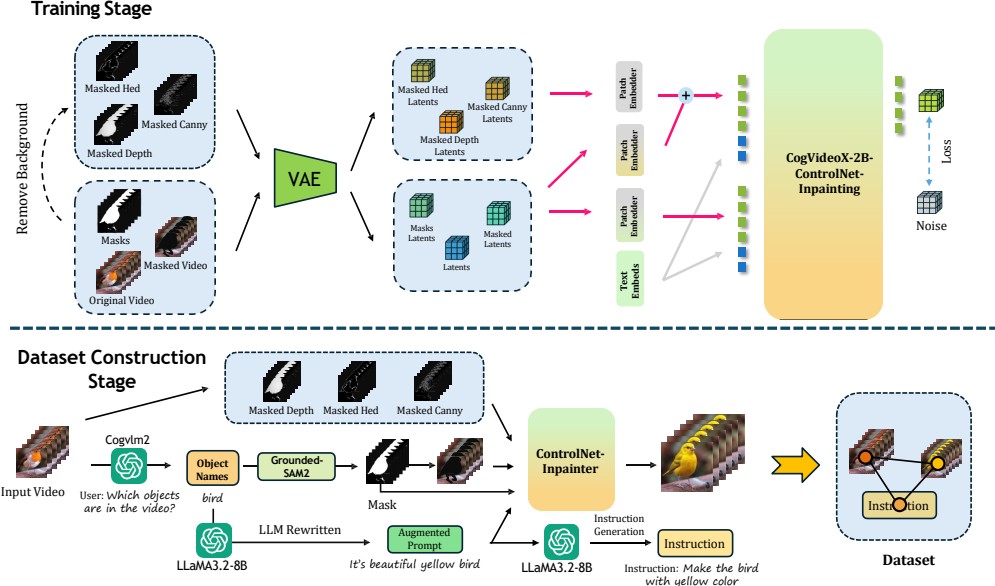

Figure 9: The training pipeline of our local stylizer. The local stylizer uses the concatenation of a noisy latents, masks latents, and masked latents as the input in the main branch. The shape of the input is $3c \times f \times h \times w$. For condition module, it takes masked HED, Canny, and depth latents as the input in the control branch with the shape of $3c \times f \times h \times w$. Our training target is to reconstruct the original latents. In this way, in the inference stage, we can change the attributes of the object via giving text instruction.

**Architectures**. We use the same controlnet architecture as in our global stylizer A.2. The difference between two models mainly lies in the base model and input condition. For our local stylizer, we utilize the CogVideoX-2B model as the base. As shown in Figure 9, the main branch takes the original video latents, masked latents, and masks latents as input (48 channels). To mitigate the inflated channel dimension, we initialize our patch embedder using the first 16 channels from CogVideoX-2B, while the remaining 32 channels are zero-initialized. Similarly, the patch embedder for control branch are also zero-initialized. Our control branch consists of 6 DiT blocks copied from main branch.

**Training Details**. We trained our local stylizer for 1 epoch, with a batch size of 32, AdamW optimizer [33], a learning rate of 1e-5, and a weight decay of 1e-4. The training videos consist of 33 frames at a resolution of $336 \times 592$. Similarly, to preserve generalization ability and accelerate training, we freeze the FFN layers except for the first DiT block.

**Inference Details**. For inference, we use a classifier-free guidance scale of 6. The inference process completes within 1 minute on 4090 GPU for a video with a resolution of $336 \times 592$, 33 frames.

**Evaluation.** Table 8 presents a quantitative comparison of local stylization methods. Our expert model achieves the lowest Ewarp score of 6.50, indicating minimal warping artifacts, and the highest CLIPScore of 0.2944, demonstrating strong semantic alignment with the input text. It also attains the best temporal consistency score of 0.9828, ensuring smooth transitions across frames. In terms of background preservation, our model surpasses all baselines by achieving the highest PSNR of 28.29, reflecting superior structural fidelity to the original background. While InsV2V exhibits competitive results, it falls short in both CLIPScore and background preservation. Overall, these results highlight the effectiveness of our model in balancing high-quality stylization with robust background consistency. Note that VACE repaints only the object itself without preserving structural information; therefore, we compare our expert model only against general editing methods.

## A.4 The Construction of Text-Guided Video Inpainter

Although many studies have explored text-guided video inpainting, such as AVID [53] and CO-COCO [53], most of these methods rely on outdated video foundation models, such as AnimateD-

*Paint the purple flower pink color.*

*Paint this cat's fur purple.*

*Make the controller pink.*

*Make the grass green.*

*Make the tree green.*

*Make the candle golden.*

*Make the ferris wheel golden.*

*Make the mask metallic color.*

*Make the headset pink.*

Figure 10: The visual results of our local stylizer. The video on the left depicts the original video, while the video on the right displays the edited videos. *Best viewed with Acrobat Reader. Click the images to play the animation clips.*

Table 9: Quantitative Comparison on Object Swap. Temp-Cons is the temporal consistency. The best results are **boldfaced**.

| Methods | Ewarp $(10^{-3})$ ($\downarrow$) | CLIPScore ($\uparrow$) | Temp-Cons ($\uparrow$) | PSNR ($\uparrow$) | SSIM ($\uparrow$) |
|---|---|---|---|---|---|
| Tokenflow [13] | 17.21 | 0.3028 | 0.9752 | 18.70 | 0.8569 |
| Flatten [9] | 17.91 | 0.2223 | 0.9744 | 18.80 | 0.8572 |
| InsV2V [8] | **8.80** | 0.2733 | 0.9722 | 21.57 | 0.9204 |
| AnyV2V [24] | 13.49 | 0.2870 | 0.9741 | 19.78 | 0.8903 |
| VACE [21] | 11.40 | 0.2860 | **0.9807** | 21.06 | 0.9170 |
| Our Expert | 12.06 | **0.3186** | 0.9782 | **25.59** | **0.9620** |

iff [16]. Consequently, the generated videos often exhibit noticeable artifacts and inconsistencies. Recently, Hu et al. proposed the VIVID model, which trains an inpainter based on CogVideoX-5B-I2V. Unfortunately, their inpainter has not been open-sourced. Similar with the VIVID [18], we use the same architecture, the first-frame edited by a stable and well-performed image editor Flux-Fill to guide the inpainting process.

**Training Details**. For training, we employed a diverse set of mask shapes to reduce the risk of our inpainter over-fitting to any particular mask geometry. At initialization, the first frame of every mask sequence was set to zeros, letting the network leverage the edited image itself as guidance,

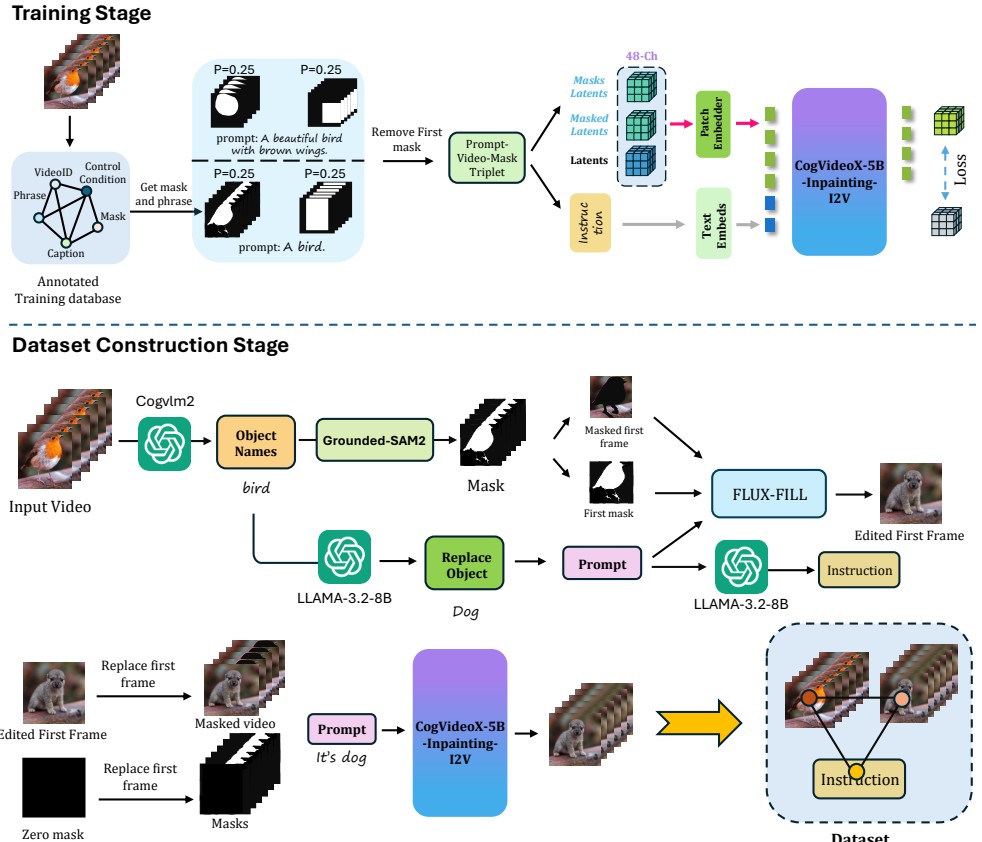

Figure 11: The training pipeline of our inpainter. Our inpainter employs a first-frame guided training strategy, where we randomly sample a mask from four different types during training. The input to DiT's patch embedder concatenates the noisy latents, masks latents, and masked latents into a $3c \times f \times h \times w$ tensor. In training, the first frame of the masked video is taken directly from the original video, and the rest are holed frames. The first frame of masks is all zeros and the rest are holed masks. During inference, the first frame of the masked video is replaced by a Flux-Fill-inpainted image, the first frame of masks are all zeros, and subsequent frames use dilated holed masks.

and the inpainter's weights were seeded from CogVideoX-5B-I2V. Because our method inpaints locally—without the auxiliary control branch required by global stylizers—we could adopt a larger batch size. Concretely, we trained the model for one epoch on our expert dataset using the AdamW optimizer [33] with a batch size of 16 and a learning rate of $1 \times 10^{-5}$. All experiments were conducted at a resolution of $336 \times 592$ over 33 frames with a temporal stride of 2. During fine-tuning, every FFN layer was frozen except that in the first DiT block, enabling efficient adaptation while preserving the backbone's capacity.

**Inference Details**. During inference, we input the prepared prompts, dilated precise masks, and videos to generate the inpainted video. The first frame is edited by Flux-Fill with a new object name. The new object name are generated by LLM. We use the classifier-free guidance of 6. The inference process can be finished on an 4090 GPU within 2 minutes, 33 frames and resolution of $336 \times 592$.

**Evaluations**. Table 9 compares different methods for object swapping. Our expert model achieves the highest CLIPScore, PNSR and SSIM, indicating strong alignment with textual instructions and background preservation. Although InsV2V reports the lowest Ewarp value, suggesting fewer warping artifacts, it performs poorly in terms of CLIPScore, reflecting weak alignment with the given text. This discrepancy stems from InsV2V's frequent failure to follow instructions and complete the object swap. Consequently, many failure cases closely resemble the original video, resulting in a lower warping error but also a significantly reduced CLIPScore. Our concurrent work, VACE,

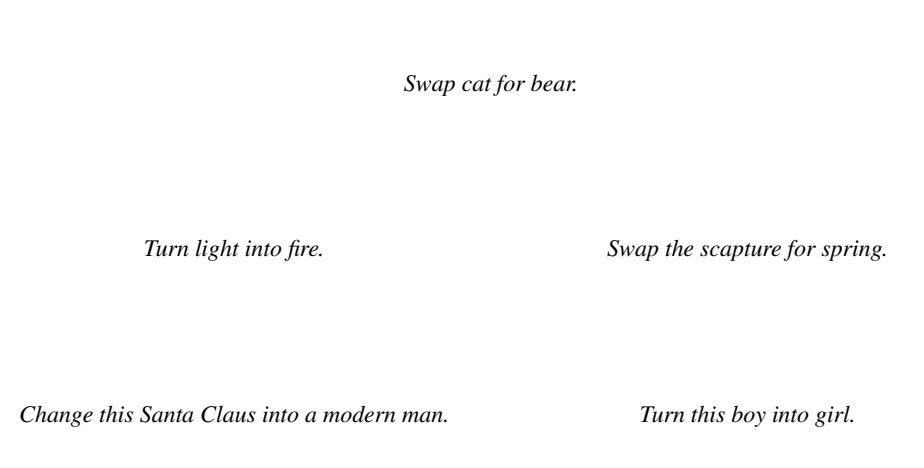

*Swap cat for bear.*

*Turn light into fire.*        *Swap the scapture for spring.*

*Change this Santa Claus into a modern man.*       *Turn this boy into girl.*

*Turn strawberry into orange.*       *Turn butterfly into seven-spot ladybird.*

Figure 12: The visual results of our inpainter. The video on the left depicts the original video, while the video on the right displays the edited videos. *Best viewed with Acrobat Reader. Click the images to play the animation clips.*

achieves the best temporal consistency, slightly outperforming our method. However, its lower CLIPScore indicates weak text alignment capability, which results in its lost opportunity to be used for creating object swap samples. These findings demonstrate that our model effectively balances object replacement precision with background preservation quality.

### A.5 The Construction of Remover

Traditional video inpainter, such as Propainter [55], uses optical flow to guide the completion. However, these methods show weaker performance than diffusion model [25]. Inpainters, such as COCOCO [56], AVID [53] are designed to add objects. Based on removal results of current video remover[25, 14, 2, 21, 27], we found that the model tends to regenerate objects in the given masked region, referring to the mask shape. To overcome this drawback, we design a training paradigm to break the correlation between generated content and mask shape.

**Mask Selection**. As shown in Figure 13, our remover is trained by assuming that the input video contains objects from unrelated videos. The model is provided with an arbitrary mask from another

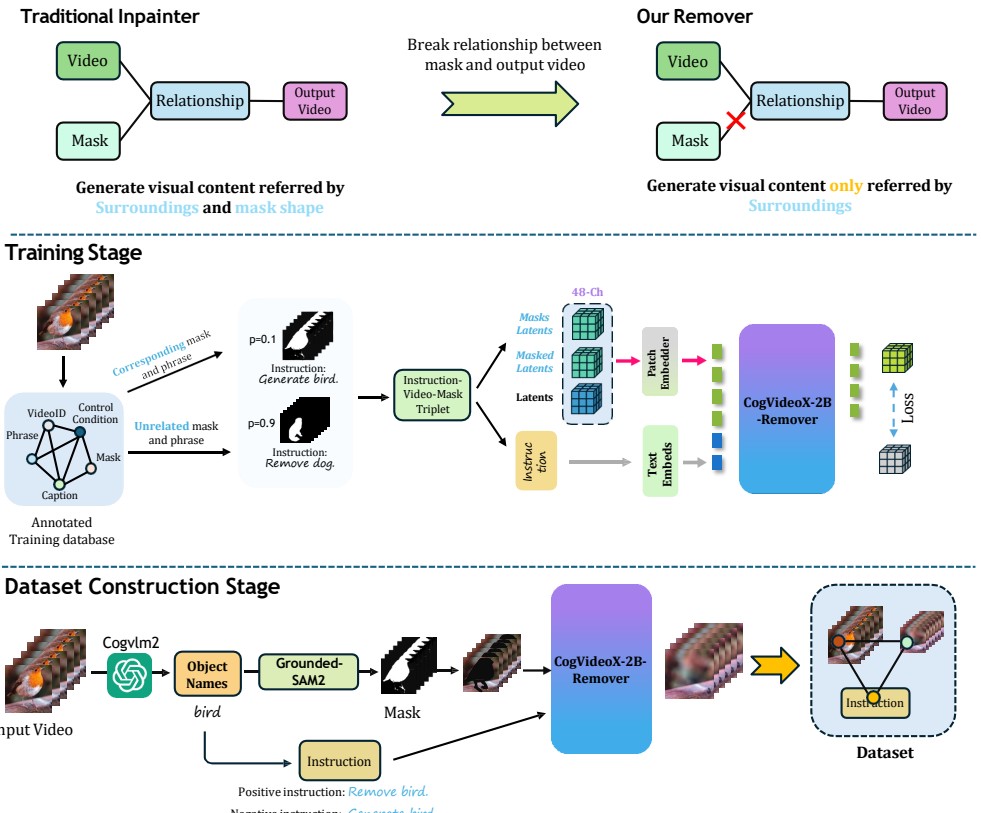

Figure 13: The training pipeline of our remover. The remover is trained with 90% unrelated masks and 10% related ones. Input to the patch embedder includes latents (noisy latents), masked latents, and masks latents, its shape is $3c \times f \times h \times w$. In the training stage, our training pipeline encourages the model to remove the object by using positive instructions (analogous to the conditional generation in CFG), and discourages the model to generate the object by using negative instructions (analogous to the unconditional generation in CFG). In the inference stage, we enable model to perform the object removal with CFG, via using positive and negative instructions.

Table 10: Quantitative Comparison of Object Removal. To assess the performance of object removal, we calculate the CLIP similarity between the removal instruction and the edited video, denoted as **relevance**. A lower relevance score indicates better removal performance. Temp-Cons is the temporal consistency. The best results are **blodfaced**.

| Methods | Ewarp($10^{-3}$) ($\downarrow$) | Relevance ($\downarrow$) | Temp-Cons ($\uparrow$) | PSNR ($\uparrow$) | SSIM ($\uparrow$) |
|---|---|---|---|---|---|
| Tokenflow [13] | 16.34 | 0.1597 | 0.9786 | 18.38 | 0.8395 |
| Flatten [9] | 11.18 | 0.2194 | 0.9759 | 18.87 | 0.8367 |
| InsV2V [8] | 6.67 | 0.2134 | 0.9747 | 22.27 | 0.9187 |
| AnyV2V [24] | 13.14 | 0.1774 | 0.9765 | 19.80 | 0.8825 |
| ProPainter [55] | 4.93 | 0.1685 | 0.9862 | **36.87** | **0.9978** |
| FloED [14] | 5.27 | 0.1920 | 0.9784 | 21.75 | 0.8948 |
| VideoPainter [2] | 8.71 | 0.1942 | 0.9787 | 26.65 | 0.9748 |
| VACE [21] | 8.65 | 0.1688 | 0.9816 | 19.59 | 0.8688 |
| Our Expert | **4.21** | **0.1554** | **0.9864** | 29.16 | 0.9863 |

video and learns to remove the assumed object while generating the object in the input video. Specifically, we randomly sample a mask and phrase from other video and used this mask to remove regions from the given video. We take 90% unrelated masks with instruction "*Remove {object name}*",

*Remove the flower.*

*Remove the cup.*                    *Remove the man.*

*Remove the man.*                    *Remove the girl.*

*Remove the giraffe.*                *Remove the seagull.*

*Remove the cup.*                    *Remove the man.*

Figure 14: The visual results of our object remover. The video on the left depicts the original video, while the video on the right displays the edited videos. *Best viewed with Acrobat Reader. Click the images to play the animation clips.*

and 10% masks corresponding to the input videos with instruction "*Generate {object name}*" This can be viewed as we use mask and the generate instruction corresponding to the input video as negative condition. During inference, the classifier-free guidance will steer the generation away from the negative condition, thus achieving the object removal.

**Training Details**. We train the remover on our expert dataset for 1 epoch with AdamW optimizer, a batch size of 32, a learning rate of 1e-5, and a weight decay of 1e-4. For data sampling, we selected 90% of the samples as task-irrelevant masks and 10% as task-relevant masks. The video was sampled at 33 frames with a stride of 2, and the resolution was set to $336 \times 592$. Our Remover is built upon the CogVideoX-2B model and initialized with its pre-trained parameters. Similarly, to preserve generalization ability and accelerate training, we freeze the FFN layers except for the first DiT block.

**Inference Details**. During inference, we use classifier-free guidance scale of 2, the positive prompt is "*Remove {object name}*", while the negative prompt is "*Generate {object name}*". The frame number is 33 and the resolution of $336 \times 592$. The removal process can be finished within 1 minute on a 4090 GPU.

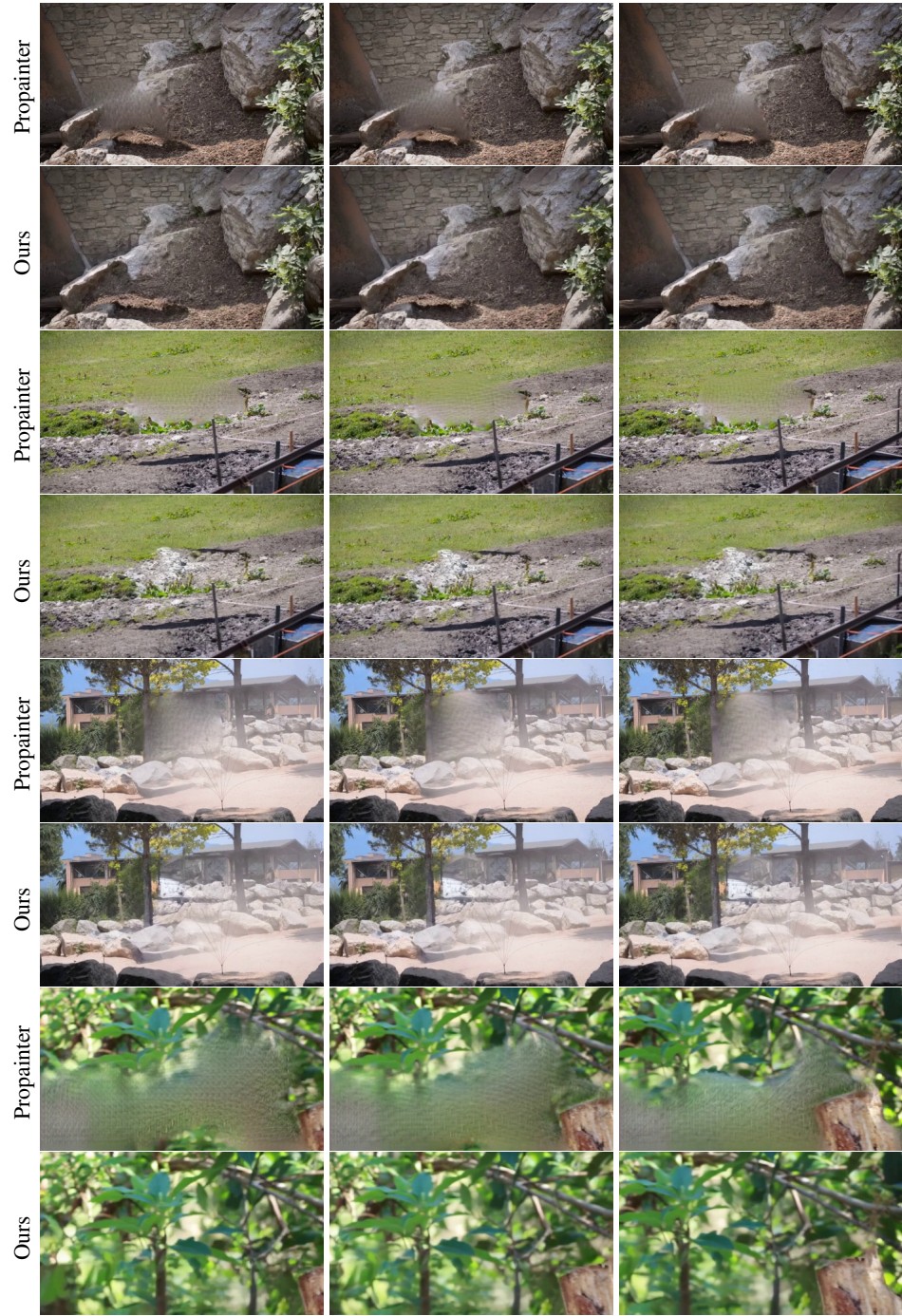

Figure 15: Removal results compared between our expert and Propainter.

**Evaluations**. Table 10 compares different methods for object removal. Our expert model achieves the lowest Ewarp score of 4.21 and a Relevance score of 0.1554, indicating minimal warping artifacts and strong removal effectiveness. It also achieves the highest Temporal Consistency of 0.9864, ensuring smooth and stable object removal across frames. Although ProPainter reports exceptionally high PSNR of 36.87 and SSIM of 0.9978, this is mainly because it preserves background pixels without alteration. However, its object removal quality is poor, as the removed regions appear significantly blurry, as shown in the qualitative examples in Figure 15. In contrast, our model effectively balances object removal and background preservation, maintaining both high visual quality and semantic alignment.

# B Style Prompts

---

**Style Prompts**

Michelangelo, Michelangelo style, Renaissance style, the art created by Michelangelo

Monet, Monet style, Impressionist style, the art created by Monet

Paul Cézanne, Cézanne style, Post-Impressionist style, the art created by Paul Cézanne

Mark Rothko, Rothko style, Abstract Expressionist style, the art created by Mark Rothko

Paul Klee, Klee style, Abstract style, Bauhaus style, the art created by Paul Klee

Picasso, Picasso style, Cubist style, the art created by Picasso

Piet Mondrian, Mondrian style, De Stijl style, the art created by Piet Mondrian

Pierre-Auguste Renoir, Renoir style, Impressionist style, the art created by Pierre-Auguste Renoir

Rembrandt, Rembrandt style, Baroque style, the art created by Rembrandt

René Magritte, Magritte style, Surrealist style, the art created by René Magritte

Roy Lichtenstein, Lichtenstein style, Pop Art style, the art created by Roy Lichtenstein

Salvador Dalí, Dalí style, Surrealist style, the art created by Salvador Dalí

Sandro Botticelli, Botticelli style, Early Renaissance style, the art created by Sandro Botticelli

Takashi Murakami, Murakami style, Superflat style, the art created by Takashi Murakami

Van Gogh, Van Gogh style, Post-Impressionist style, the oil painting style, the oil painting created by Van Gogh

Wassily Kandinsky, Kandinsky style, Abstract style, Bauhaus style, the art created by Wassily Kandinsky

Mat Collishaw, Collishaw style, Contemporary Art style, the art created by Mat Collishaw

Yayoi Kusama, Kusama style, Contemporary Art style, Pop Art style, the art created by Yayoi Kusama

Igor Morski, Morski style, Surrealist style, Fantasy Art style, the art created by Igor Morski

Shinkai Makoto, Shinkai style, Anime style, Cinematic style, the art created by Shinkai Makoto

Pixar, Pixar style, 3D Animation style, CGI style, the animation created by Pixar

Kyoto Animation, Kyoto Animation style, Anime style, the animation created by Kyoto Animation

Jerry Pinkney, Pinkney style, Illustration style, Children's Books style, the illustrations created by Jerry Pinkney

Hayao Miyazaki, Miyazaki style, Anime style, Ghibli style, the animation created by Hayao Miyazaki

Beatrix Potter, Potter style, Illustration style, Children's Books style, the illustrations created by Beatrix Potter

Jon Klassen, Klassen style, Children's Books style, Illustration style, the illustrations created by Jon Klassen

Kay Sage, Sage style, Surrealist style, the art created by Kay Sage

Jeffrey Catherine Jones, Jones style, Fantasy Art style, Illustration style, the art created by Jeffrey Catherine Jones

Yaacov Agam, Agam style, Kinetic Art style, Op Art style, the art created by Yaacov Agam

David Hockney, Hockney style, Pop Art style, Contemporary Art style, the art created by David Hockney

Victor Moscoso, Moscoso style, Psychedelic Art style, Graphic Art style, the art created by Victor Moscoso

Raphaelite, Pre-Raphaelite Brotherhood style, the art created by Raphaelite

Stefan Koid, Koid style, Contemporary Art style, the art created by Stefan Koid

Sui Ishida, Ishida style, Manga style, the art created by Sui Ishida

John Harris, Harris style, Sci-Fi Art style, Illustration style, the art created by John Harris

Jon Klassen, Klassen style, Children's Books style, Illustration style, the art created by Jon Klassen

Junji Ito, Ito style, Horror Manga style, the art created by Junji Ito Koe no Katachi, Koe no Katachi style, Anime style, Manga style, the art created in Koe no Katachi style

---

## Style Prompts

Anton Pieck, Pieck style, Illustration style, Fairy Tale Art style, the art created by Anton Pieck

Carl Barks, Barks style, Comic Book Art style, the art created by Carl Barks

Alphonse Mucha, Mucha style, Art Nouveau style, the art created by Alphonse Mucha

Andy Warhol, Warhol style, Pop Art style, the art created by Andy Warhol

Banksy, Banksy style, Street Art style, Contemporary Art style, the art created by Banksy

Francisco de Goya, Goya style, Romanticism style, the art created by Francisco de Goya

Caravaggio, Caravaggio style, Baroque style, the art created by Caravaggio

Diego Rivera, Rivera style, Muralism style, the art created by Diego Rivera

Marc Chagall, Chagall style, Modern Art style, Surrealist style, the art created by Marc Chagall

Edgar Degas, Degas style, Impressionist style, the art created by Edgar Degas

Eugène Delacroix, Delacroix style, Romanticism style, the art created by Eugène Delacroix

Francis Bacon, Bacon style, Expressionist style, Modern Art style, the art created by Francis Bacon

Frida Kahlo, Kahlo style, Surrealist style, Modern Art style, the art created by Frida Kahlo

Gerald Brom, Brom style, Dark Fantasy Art style, the art created by Gerald Brom

Gustav Klimt, Klimt style, Symbolist style, Art Nouveau style, the art created by Gustav Klimt

Henri Matisse, Matisse style, Fauvist style, the art created by Henri Matisse

J.M.W. Turner, Turner style, Romanticism style, the art created by J.M.W. Turner

Jack Kirby, Kirby style, Comic Book Art style, the art created by Jack Kirby

Jackson Pollock, Pollock style, Abstract Expressionist style, the art created by Jackson Pollock

Johannes Vermeer, Vermeer style, Baroque style, the art created by Johannes Vermeer

Jean-Michel Basquiat, Basquiat style, Neo-Expressionist style, the art created by Jean-Michel Basquiat

Marcel Duchamp, Duchamp style, Dada style, the art created by Marcel Duchamp

Traditional Chinese Ink Painting, Chinese Art style, the art created in Traditional Chinese Ink Painting style

Japanese Ukiyo-e, Ukiyo-e style, Japanese Art style, the art created in Japanese Ukiyo-e style

Japanese comics/manga, Manga style, the art created in Japanese comics/manga style

Stock illustration style, Illustration style, the illustrations created in Stock illustration style

CGSociety, CGSociety style, Digital Art style, CGI style, the art created by CGSociety

DreamWorks Pictures, DreamWorks style, 3D Animation style, CGI style, the animation created by DreamWorks Pictures

Fashion, Fashion Illustration style, Runway Art style, the art created in Fashion style

Poster of Japanese graphic design, Japanese Graphic Design style, the art created in Poster of Japanese graphic design style

90s video game, Retro Game Art style, the art created in 90s video game style

French art, Various Styles (Impressionism, Romanticism, etc.), the art created in French art style

Bauhaus, Bauhaus style, Modernist Art style, the art created in Bauhaus style

Anime, Anime style, Japanese Animation style, the art created in Anime style

Pixel Art, Pixel Art style, Digital Art style, Retro Art style, the art created in Pixel Art style

Vintage, Vintage style, Retro Art style, the art created in Vintage style

Pulp Noir, Pulp Noir style, Pulp Art style, Noir Art style, the art created in Pulp Noir style

Country style, Folk Art style, the art created in Country style

Abstract, Abstract style, Abstract Art style, the art created in Abstract style

Risograph, Risograph style, Printmaking style, Graphic Art style, the art created in Risograph style

Graphic, Graphic style, Graphic Design style, the art created in Graphic style

Ink render, Ink render style, Ink Art style, the art created in Ink render style

Ethnic Art, Ethnic Art style, Folk Art style, the art created in Ethnic Art style

Retro dark vintage, Retro dark vintage style, Gothic Art style, Dark Art style, the art created in Retro dark vintage style

## Style Prompts

Gothic gloomy, Gothic gloomy style, Gothic Art style, the art created in Gothic gloomy style
Realism, Realism style, Realist Art style, the art created in Realism style
Black and white, Black and white style, Monochrome Art style, the art created in Black and white style
Unity Creations, Unity Creations style, Digital Art style, CGI style, the art created by Unity Creations
Baroque, Baroque style, Baroque Art style, the art created in Baroque style
Impressionism, Impressionist style, the art created in Impressionism style
Art Nouveau, Art Nouveau style, the art created in Art Nouveau style Rococo, Rococo style, Rococo Art style, the art created in Rococo style
Adrian Donohue, Donohue style, Photography style, the photographs taken by Adrian Donohue
Adrian Tomine, Tomine style, Comic Art style, Illustration style, the art created by Adrian Tomine
Akihiko Yoshida, Yoshida style, Video Game Art style, Concept Art style, the art created by Akihiko Yoshida
Akira Toriyama, Toriyama style, Manga style, Anime style, the art created by Akira Toriyama
Cai Guo-Qiang, Cai style, Contemporary Art style, the art created by Cai Guo-Qiang
Drew Struzan, Struzan style, Poster Art style, Illustration style, the art created by Drew Struzan
Hans Arp, Arp style, Dada style, Abstract Art style, the art created by Hans Arp
Ilya Kuvshinov, Kuvshinov style, Manga style, Anime style, the art created by Ilya Kuvshinov
James Jean, Jean style, Illustration style, Fine Art style, the art created by James Jean
Jasmine Becket-Griffith, Becket-Griffith style, Pop Surrealism style, the art created by Jasmine Becket-Griffith
Jean Giraud, Giraud style, Comic Art style, Illustration style, the art created by Jean Giraud
Partial anatomy, Anatomical Art style, the art created in Partial anatomy style
Color ink on paper, Ink Art style, the art created with color ink on paper
Doodle, Doodle style, Illustration style, Sketch Art style, the art created in Doodle style
Voynich manuscript, Manuscript Art style, Historical Art style, the art created in Voynich manuscript style
Book page, Book page style, Illustration style, Typography Art style, the art created in Book page style
Realistic, Realism style, the art created in Realistic style
3D, 3D Art style, CGI style, the art created in 3D style
Sophisticated, Fine Art style, the art created in Sophisticated style
Photoreal, Photorealism style, the art created in Photoreal style
Character concept art, Character concept art style, Concept Art style, the art created in Character concept art style
Renaissance, Renaissance style, Renaissance Art style, the art created in Renaissance style
Fauvism, Fauvist style, the art created in Fauvism style
Cubism, Cubist style, the art created in Cubism style
Abstract Art, Abstract Art style, the art created in Abstract Art style
Surrealism, Surrealist style, the art created in Surrealism style
Op Art / Optical Art, Optical Art style, the art created in Op Art / Optical Art style
Victorian, Victorian style, Victorian Art style, the art created in Victorian style
Futuristic, Futuristic style, Sci-Fi Art style, the art created in Futuristic style
Minimalist, Minimalist style, the art created in Minimalist style
Brutalist, Brutalist style, the art created in Brutalist style
Constructivist, Constructivist style, the art created in Constructivist style
BOTW, BOTW style, Video Game Art style (Breath of the Wild), the art created in BOTW style
Warframe, Warframe style, Video Game Art style, the art created in Warframe style
Pokémon, Pokémon style, Anime style, Video Game Art style, the art created in Pokémon style

## Style Prompts

CookieRun Kingdom, CookieRun Kingdom style, Video Game Art style, the art created in CookieRun Kingdom style

League of Legends, League of Legends style, Video Game Art style, the art created in League of Legends style

Jojo's Bizarre Adventure, Jojo's Bizarre Adventure style, Manga style, Anime style, the art created in Jojo's Bizarre Adventure style

Makoto Shinkai, Shinkai style, Anime style, Cinematic style, the art created by Makoto Shinkai

Poster of Japanese graphic design, Japanese Graphic Design style, the art created in Poster of Japanese graphic design style

90s video game, Retro Game Art style, the art created in 90s video game style

French art, Various Styles (Impressionism, Romanticism, etc.), the art created in French art style

Bauhaus, Bauhaus style, Modernist Art style, the art created in Bauhaus style

Anime, Anime style, Japanese Animation style, the art created in Anime style

Pixel Art, Pixel Art style, Digital Art style, Retro Art style, the art created in Pixel Art style

Vintage, Vintage style, Retro Art style, the art created in Vintage style

Pulp Noir, Pulp Noir style, Pulp Art style, Noir Art style, the art created in Pulp Noir style

Country style, Folk Art style, the art created in Country style

Abstract, Abstract style, Abstract Art style, the art created in Abstract style

Risograph, Risograph style, Printmaking style, Graphic Art style, the art created in Risograph style

Graphic, Graphic style, Graphic Design style, the art created in Graphic style

Ink render, Ink render style, Ink Art style, the art created in Ink render style

Ethnic Art, Ethnic Art style, Folk Art style, the art created in Ethnic Art style

Retro dark vintage, Retro dark vintage style, Gothic Art style, Dark Art style, the art created in Retro dark vintage style

Traditional Chinese Ink Painting style, Chinese Art style, the art created in Traditional Chinese Ink Painting style

Steampunk, Steampunk style, Steampunk Art style, the art created in Steampunk style

Film photography, Film photography style, Photography style, the photographs taken in Film photography style

Concept art, Concept art style, Conceptual Art style, the art created in Concept art style

Montage, Montage style, Collage Art style, the art created in Montage style

Full details, Full details style, Realism style, Hyperrealism style, the art created in Full details style

Gothic gloomy, Gothic gloomy style, Gothic Art style, the art created in Gothic gloomy style

Realism, Realism style, Realist Art style, the art created in Realism style

Black and white, Black and white style, Monochrome Art style, the art created in Black and white style

Unity Creations, Unity Creations style, Digital Art style, CGI style, the art created by Unity Creations

Baroque, Baroque style, Baroque Art style, the art created in Baroque style

Impressionism, Impressionist style, the art created in Impressionism style

Art Nouveau, Art Nouveau style, the art created in Art Nouveau style

Rococo, Rococo style, Rococo Art style, the art created in Rococo style

Adrian Donohue, Donohue style, Photography style, the photographs taken by Adrian Donohue

Adrian Tomine, Tomine style, Comic Art style, Illustration style, the art created by Adrian Tomine

Akihiko Yoshida, Yoshida style, Video Game Art style, Concept Art style, the art created by Akihiko Yoshida

## Style Prompts

Book page, Book page style, Illustration style, Typography Art style, the art created in Book page style

Realistic, Realism style, the art created in Realistic style

3D, 3D Art style, CGI style, the art created in 3D style

Sophisticated, Fine Art style, the art created in Sophisticated style

Photoreal, Photorealism style, the art created in Photoreal style

Character concept art, Character concept art style, Concept Art style, the art created in Character concept art style

Renaissance, Renaissance style, Renaissance Art style, the art created in Renaissance style

Fauvism, Fauvist style, the art created in Fauvism style

Cubism, Cubist style, the art created in Cubism style

Abstract Art, Abstract Art style, the art created in Abstract Art style Surrealism, Surrealist style, the art created in Surrealism style

Op Art / Optical Art, Optical Art style, the art created in Op Art / Optical Art style

Futuristic, Futuristic style, Sci-Fi Art style, the art created in Futuristic style

Minimalist, Minimalist style, the art created in Minimalist style

Brutalist, Brutalist style, the art created in Brutalist style

Constructivist, Constructivist style, the art created in Constructivist style

BOTW, BOTW style, Video Game Art style (Breath of the Wild), the art created in BOTW style

Warframe, Warframe style, Video Game Art style, the art created in Warframe style

Pokémon, Pokémon style, Anime style, Video Game Art style, the art created in Pokémon style

APEX, APEX style, Video Game Art style, the art created in APEX style

The Elder Scrolls, Elder Scrolls style, Video Game Art style, the art created in The Elder Scrolls style

From Software, From Software style, Video Game Art style, the art created by From Software

Detroit: Become Human, Detroit: Become Human style, Video Game Art style, the art created in Detroit: Become Human style

AFK Arena, AFK Arena style, Video Game Art style, the art created in AFK Arena style

CookieRun Kingdom, CookieRun Kingdom style, Video Game Art style, the art created in CookieRun Kingdom style

League of Legends, League of Legends style, Video Game Art style, the art created in League of Legends style

Jojo's Bizarre Adventure, Jojo's Bizarre Adventure style, Manga style, Anime style, the art created in Jojo's Bizarre Adventure style

Makoto Shinkai, Shinkai style, Anime style, Cinematic style, the art created by Makoto Shinkai

Soejima Shigenori, Shigenori style, Video Game Art style, the art created by Soejima Shigenori

Yamada Akihiro, Akihiro style, Manga style, Anime style, the art created by Yamada Akihiro

Munashichi, Munashichi style, Concept Art style, Digital Art style, the art created by Munashichi

Watercolor Children's Illustration, Watercolor Children's Illustration style, Watercolor Art style, Children's Books style, the art created in Watercolor Children's Illustration style

Ghibli Studio, Ghibli style, Anime style, the animation created by Ghibli Studio

Stained Glass Window, Stained Glass style, the art created in Stained Glass Window style

Ink Illustration, Ink Illustration style, Ink Art style, the art created in Ink Illustration style

Miyazaki Hayao Style, Miyazaki style, Anime style, Ghibli style, the animation created in Miyazaki Hayao style

Vincent van Gogh, Van Gogh style, Post-Impressionist style, the oil painting style, the oil painting created by Van Gogh

Leonardo da Vinci, Da Vinci style, Renaissance style, the art created by Leonardo da Vinci

Manga, Manga style, the art created in Manga style

Pointillism, Pointillist style, the art created in Pointillism style

Claude Monet, Monet style, Impressionist style, the art created by Claude Monet

Johannes Itten, Itten style, Bauhaus style, the art created by Johannes Itten

## Style Prompts

Osamu Tezuka, Tezuka style, Manga style, Anime style, the art created by Osamu Tezuka

Rob Gonsalves, Gonsalves style, Surrealist style, Magic Realism style, the art created by Rob Gonsalves

Sol LeWitt, LeWitt style, Minimalist style, Conceptual Art style, the art created by Sol LeWitt

Yusuke Murata, Murata style, Manga style, Anime style, the art created by Yusuke Murata

Antonio Mora, Mora style, Surrealist style, Photo Manipulation style, the art created by Antonio Mora

Yoji Shinkawa, Shinkawa style, Video Game Art style, Concept Art style, the art created by Yoji Shinkawa

National Geographic, National Geographic style, Photography style, the photographs taken for National Geographic

Hyperrealism, Hyperrealism style, the art created in Hyperrealism style Cinematic, Cinematic style, Cinematic Art style, the art created in Cinematic style

Architectural Sketching, Architectural Sketching style, Architecture Art style, the art created in Architectural Sketching style

Clear Facial Features, Clear Facial Features style, Portrait Art style, the art created with Clear Facial Features

Interior Design, Interior Design style, Interior Art style, the art created in Interior Design style

Weapon Design, Weapon Design style, Concept Art style, the art created in Weapon Design style

Subsurface Scattering, Subsurface Scattering style, Digital Art style, CGI style, the art created with Subsurface Scattering

Game Scene Graph, Game Scene Graph style, Video Game Art style, the art created in Game Scene Graph style

Cyberpunk style, neon-lit dystopian cityscape, futuristic skyscrapers, dark rain-soaked streets, glowing holograms, cybernetic characters, high-tech and gritty, rebellion theme, vibrant colors, immersive detail

Ultra-detailed photorealistic image, realistic lighting and textures, high-resolution, cinematic quality

Cyberpunk style, neon-lit cityscape, futuristic tech, dark atmosphere, glowing holograms, high-tech low-life

Epic fantasy scene, magical landscapes, mythical creatures, intricate details, vibrant colors, ethereal lighting

Anime-style illustration, vibrant colors, dynamic poses, sharp line art, expressive characters, 2D aesthetics

Concept art, highly detailed environments, creative landscapes, futuristic design, cinematic lighting, imaginative visuals

Watercolor painting, soft textures, pastel colors, flowing brushstrokes, dreamy and artistic

Steampunk aesthetic, Victorian-era technology, brass and gears, intricate machinery, retro-futuristic design

Abstract art, vibrant colors, geometric patterns, fluid forms, modern artistic expression, minimalistic or chaotic

Noir style, black-and-white, dramatic shadows, moody atmosphere, vintage detective aesthetics

Pixel art, retro 8-bit style, vibrant blocky colors, low-resolution, game-like visuals, nostalgic charm

Professional studio portrait, dramatic lighting, high detail, shallow depth of field, realistic skin textures

Isometric perspective, detailed environments, vibrant colors, 3D-inspired flat design, intricate details

Baroque art, elaborate and ornate details, dramatic compositions, rich textures, classical European aesthetics

Dark fantasy setting, eerie atmosphere, mystical creatures, gothic architecture, muted tones, ominous lighting

## Style Prompts

Full details, Full details style, Realism style, Hyperrealism style, the art created in Full details style

Chibi-style characters, exaggerated cute proportions, vibrant colors, anime-inspired, playful and adorable

Dennis Stock, Stock style, Photography style, the photographs taken by Dennis Stock

Michal Lisowski, Lisowski style, Digital Art style, Illustration style, the art created by Michal Lisowski

Paul Lehr, Lehr style, Science Fiction Art style, Illustration style, the art created by Paul Lehr

Ross Tran, Tran style, Digital Art style, Concept Art style, the art created by Ross Tran

Montage, Montage style, Collage Art style, the art created in Montage style

Swoon, Swoon style, Street Art style, Contemporary Art style, the art created by Swoon

Tasha Tudor, Tudor style, Illustration style, Children's Books style, the illustrations created by Tasha Tudor

Tintoretto, Tintoretto style, Mannerism style, Late Renaissance style, the art created by Tintoretto

Theodore Robinson, Robinson style, Impressionist style, the art created by Theodore Robinson

Titian, Titian style, Renaissance style, the art created by Titian

WLOP, WLOP style, Digital Art style, Fantasy Art style, the art created by WLOP

Yanjun Cheng, Cheng style, Contemporary Art style, the art created by Yanjun Cheng

Yoji Shinkawa, Shinkawa style, Video Game Art style, Concept Art style, the art created by Yoji Shinkawa

Alena Aenami, Aenami style, Digital Art style, the art created by Alena Aenami

Anton Fadeev, Fadeev style, Concept Art style, Digital Art style, the art created by Anton Fadeev

Charlie Bowater, Bowater style, Concept Art style, Digital Art style, the art created by Charlie Bowater

Cory Loftis, Loftis style, Concept Art style, Digital Art style, the art created by Cory Loftis

Fenghua Zhong, Zhong style, Digital Art style, Illustration style, the art created by Fenghua Zhong

Greg Rutkowski, Rutkowski style, Digital Painting style, Fantasy Art style, the art created by Greg Rutkowski

Traditional Chinese Ink Painting style, Chinese Art style, the art created in Traditional Chinese Ink Painting style

Steampunk, Steampunk style, Steampunk Art style, the art created in Steampunk style

Film photography, Film photography style, Photography style, the photographs taken in Film photography style

Concept art, Concept art style, Conceptual Art style, the art created in Concept art style

Akira Toriyama, Toriyama style, Manga style, Anime style, the art created by Akira Toriyama

Cai Guo-Qiang, Cai style, Contemporary Art style, the art created by Cai Guo-Qiang

Drew Struzan, Struzan style, Poster Art style, Illustration style, the art created by Drew Struzan

Hans Arp, Arp style, Dada style, Abstract Art style, the art created by Hans Arp

Ilya Kuvshinov, Kuvshinov style, Manga style, Anime style, the art created by Ilya Kuvshinov

James Jean, Jean style, Illustration style, Fine Art style, the art created by James Jean

Jasmine Becket-Griffith, Becket-Griffith style, Pop Surrealism style, the art created by Jasmine Becket-Griffith

Jean Giraud, Giraud style, Comic Art style, Illustration style, the art created by Jean Giraud

Partial anatomy, Anatomical Art style, the art created in Partial anatomy style

Color ink on paper, Ink Art style, the art created with color ink on paper

Doodle, Doodle style, Illustration style, Sketch Art style, the art created in Doodle style

Voynich manuscript, Manuscript Art style, Historical Art style, the art created in Voynich manuscript style

Detroit: Become Human, Detroit: Become Human style, Video Game Art style, the art created in Detroit: Become Human style

AFK Arena, AFK Arena style, Video Game Art style, the art created in AFK Arena style

Hong SoonSang, SoonSang style, Animation style, Concept Art style, the art created by Hong SoonSang

> **Style Prompts**
>
> APEX, APEX style, Video Game Art style, the art created in APEX style
> The Elder Scrolls, Elder Scrolls style, Video Game Art style, the art created in The Elder Scrolls style
> From Software, From Software style, Video Game Art style, the art created by From Software
> Art Nouveau style, flowing organic lines, floral motifs, intricate patterns, pastel and earthy tones
> Cinematic lighting, moody atmosphere, dramatic shadows, high contrast, film-like quality
> Minimalist design, clean and simple lines, muted colors, open space, modern and abstract
> Retro-futurism, 1950s sci-fi style, sleek spaceships, vintage design, bold colors, nostalgic aesthetics
> Fantasy map style, hand-drawn cartography, intricate details, parchment textures, medieval aesthetic
> Glitch art, pixelated visuals, distorted and fragmented images, neon and dark tones, digital chaos
> Nature photography, high detail, realistic textures, vibrant landscapes, soft natural light
> Low poly 3D art, simplified geometric shapes, bright pastel colors, minimalist style, game aesthetic
> Impressionist painting, soft brushstrokes, vivid colors, natural light, artistic and emotional style
> Sci-fi futurism, sleek spaceships, glowing cities, alien landscapes, advanced technology, cinematic visuals
> Pop art style, bold colors, comic book aesthetics, stylized patterns, retro 1960s look
> Vaporwave style, retro-futuristic design, pastel neon colors, 1980s aesthetics, surreal landscapes
> Surrealist art, dreamlike scenes, unexpected juxtapositions, imaginative landscapes, abstract and symbolic
> Medieval-inspired style, illuminated manuscripts, intricate patterns, historical scenes, muted tones
> Graffiti art, vibrant spray paint textures, urban street style, bold typography, dynamic and expressive

## C  Limitations and Future Work

In this paper, we proposed a dataset consisting of 18 tasks with 2 million video pairs. While our method demonstrates promising editing results, the current dataset is still insufficient for training high-capacity video editors. In future work, we plan to employ more powerful expert models to generate a larger and more diverse set of video pairs, and further distill the editing capabilities from these experts into more compact and efficient models.

