# OpenReview forum: "Señorita-2M: A High-Quality Instruction-based Dataset for General Video Editing by Video Specialists"
_NeurIPS.cc/2025/Datasets_and_Benchmarks_Track — NeurIPS 2025 Datasets and Benchmarks Track poster_

### Official Review · Reviewer_EXXs · 2025-06-22

**Rating:** 4
**Confidence:** 4

**Summary:**

This submission introduces the Señorita-2M dataset, a large-scale and diverse video editing dataset designed to enhance consistency and quality in video editing tasks. The primary contribution is the development of this dataset with systematic categorization and quality-filtering processes, leading to improved model performance on video editing tasks.

**Dataset Code Accessibility:**

Yes

**Ethical Considerations:**

No, there are no or only very minor ethics concerns

**Final Justification:**

Thanks to the authors for the reply.

I respectfully disagree with the claim
> Nevertheless, editing models trained on the dataset still demonstrated robust and reliable performance, indicating that these minor flaws did not significantly impact overall effectiveness.

During the pre-training phase, foundational model requires clean and high-quality data, which has been validated across many settings, including LLM, VLM, image gen and so on. This has been reported by lots of tech reports, like the recent K2, GLM4.5...

That being said, I agree with other reviewers that this paper made a clear positive contribution to the field.

**Limitations Weaknesses:**

- The assertion of high quality in the dataset is somewhat overstated as a significant portion of videos, particularly in the object removal task, exhibit noticeable issues like strange blurry artifacts. For example, `2dbc05d7728a2e4894644a418f85386a, 7b74445894394332393b0ea4efdc0943, 8e5a5d70d675c71db1a0c317d50aefe5, 4513d753445a86c8e4d84aee6ae6ddc3, 139464ba886677abf61f0674f7f02abc`, undermining the overall quality claim. Additionally, there are instances where the edited output is entirely unsatisfactory and lacks clear interpretation, such as video `9a286a07fd0cdfee8093b969e480e6ed`.
- The fidelity of the original content preservation is questionable, as numerous videos (such as `f02506fd9fa222924d9f8176d871d085`) demonstrate unintended changes in areas that were not meant to be edited . These issues lead to a lack of consistency and reliability, which can hamper the practical usability of the dataset in developing robust video editing models.

**Strengths Contributions:**

- The Señorita-2M dataset is a good contribution to the field as it offers a large-scale and publicly accessible resource for video editing, with two million samples that are diverse in content, resolution, and frame count, addressing the scarcity of datasets in this area.
- Upon examining the dataset, it is evident that many editing outputs display a high standard of quality, particularly in maintaining the intended modifications while preserving the integrity of the surrounding frames.
- The dataset's categorization into 2 main classes and 18 subcategories provides a comprehensive structure for various video editing tasks, ensuring a wide range of scenarios for training and testing models, which is beneficial for developing more generalized video editing capabilities.

---

> ### Author Rebuttal · Authors · 2025-07-31
>
> Thank you for your comments and recognition. Your feedback, such as *"the Señorita-2M dataset is a good contribution to the field"* and *"it is evident that many editing outputs display a high standard of quality, particularly in maintaining the intended modifications while preserving the integrity of the surrounding frames"*, is really encouraging. We sincerely appreciate your time and thoughtful feedback.
>
> **Señorita-2M** is a large-scale dataset consisting of around 2 million video triplets across 18 sub-categories, covering most common tasks in the video editing. The dataset is highly diverse, featuring videos ranging from 33 to 64 frames in length, with resolutions between 336×592 and 1120×1984. It demonstrates strong generalization across various base models, including *CogVideo-5B*, *CogVideo-5B-I2V*, *Wan2.1-1.3B*, and *Wan2.1-14B-T2V*, all of which exhibit impressive editing capabilities and produce visually pleasing results. Notably, the dataset has been well received by the community, with over *15K downloads on huggingface, including 4.5K downloads in the last month*. Several recent papers have already employed Señorita-2M for training editors or developing unified generation and editing models, further highlighting its effectiveness and research value.
>
>
>
> *Overall, Señorita-2M represents a large-scale and highly diverse dataset with strong generalization across various base models. Its increasing adoption and use in recent research highlight its value to the community. We hope this work will continue to support advancements in video editing, particularly in addressing the challenges of building high-quality, general-purpose editing datasets. Once again, we sincerely appreciate your thoughtful comments and feedback. Your insights are truly valuable and will help us further enhance the quality of our manuscript.*
>
>
> &nbsp;
>
> ---
>
>
> ***According to the NeurIPS 2025 Datasets and Benchmarks track guidelines, we are not allowed to include external links or updated pdfs during the rebuttal or discussion stages. As a result, we are unable to provide additional visualization results at this time. We sincerely appreciate your understanding.***
>
>
> ---
>
> &nbsp;
>
> **Q1: The assertion of high quality in the dataset is somewhat overstated as a significant portion of videos, particularly in the object removal task, exhibit noticeable issues like strange blurry artifacts.**
>
>
> R1: Thank you for pointing out this issue. We want to clarify the following points.
>
> *  We acknowledge that our dataset contains some imperfections. Nevertheless, editing models trained on the dataset still demonstrated robust and reliable performance, indicating that these minor flaws did not significantly impact overall effectiveness.
>
> * The videos you referred to appear to be part of the *object addition* task. These videos are located in the *obj_removal_videos_upload* folder because the *source videos* were generated by an object removal model and are repurposed for the object addition task. In this setup, the **source video** is a version where an object has been removed, and the **target video** is the original unedited version. Importantly, in the object addition task, the model is trained to reconstruct the original video (**target**) from the edited version (**source**), and the **source video is not used as ground truth supervision**. Therefore, any imperfections or artifacts present in the source video do not directly impact the model’s learning or the quality of the final editing results.
>
> *   We plan to update the dataset by adding an aesthetic score [1,2] for each video. This will enable users to set a threshold to filter out low-quality videos during sampling, improving the overall data quality for specific applications.
>
> &nbsp;
>
> **Q2: The fidelity of the original content preservation is questionable, as numerous videos demonstrate unintended changes in areas that were not meant to be edited . These issues lead to a lack of consistency and reliability, which can hamper the practical usability of the dataset in developing robust video editing models.**
>
>
> R2: Thank you for your comments. We would like to provide the following clarifications:
>
> *   We acknowledge that some imperfections in our dataset. These imperfections are mainly inherited from the base model. In our paper, we focused on exploring the entire pipeline for constructing a video editing dataset, including how to build specialized editing experts and how to filter out failure cases.
>
> *   Manually annotating a large-scale video editing dataset is extremely challenging. Using tools like Adobe After Effects (AE) or Photoshop (PS) to edit each frame is not practical or scalable. As a result, the only feasible approach to obtain video editing triplets is by leveraging editing models to generate them. Given that our dataset includes 2 million video triplets, it is not realistic to manually filter every sample. Therefore, some imperfect videos are difficult to avoid.
>
> *   We are planing to use an aesthetic scoring model [1,2] to help identify and filter out failure cases. An updated version of the dataset reflecting this improvement will be released soon.
>
> [1] *LAION Aesthetic Predictor*. In github.
>
> Christoph Schuhmann.
>
> [2] *LAION-5B: An open large-scale dataset for training next generation image-text models*. In Proceedings of the Neural Information Processing Systems (NeurIPS) 2022 Datasets and Benchmarks Track.
>
> Christoph Schuhmann, Romain Beaumont, Richard Vencu, Cade Gordon, Ross Wightman, Mehdi Cherti, Theo Coombes, Aarush Katta, Clayton Mullis, Mitchell Wortsman, Patrick Schramowski, Srivatsa Kundurthy, Katherine Crowson, Ludwig Schmidt, Robert Kaczmarczyk, Jenia Jitsev.
>
>
>
>
> &nbsp;
>
> ---
>
> **We sincerely hope our response can address your concerns. Your comments are really constructive and help us improve the quality of our manuscript.**

---

### Official Review · Reviewer_pNU1 · 2025-06-23

**Rating:** 5
**Confidence:** 3

**Summary:**

This paper presents Señorita-2M, a large-scale, diverse, and high-quality instructional video editing dataset designed to address the limitations of existing datasets in terms of editing accuracy, temporal consistency, and sample diversity. The dataset systematically categorizes editing tasks into 2 major categories and 18 subcategories, leverages multiple task-specific experts for data generation, and implements content-level and instruction-level filtering pipelines to ensure data reliability. Models trained on the Señorita-2M dataset demonstrate superior visual quality, robust inter-frame consistency, and accurate instruction-following capabilities, validating the effectiveness of the proposed dataset.

**Dataset Code Accessibility:**

Partly

**Ethical Considerations:**

No, there are no or only very minor ethics concerns

**Final Justification:**

My concerns have been well addressed, so I prefer to maintain my positive score of accepting this paper. A dataset of such scale is essential to the development of generative video editing.

**Limitations Weaknesses:**

1. Imbalanced subcategories in the dataset. As illustrated in Table 2, "Object Addition" and "Object Removal" occupy 26.6% and 20.7% respectively, while some categories like "Outpainting" only constitute a small proportion at 2.0%. Such data distribution imbalance might introduce bias and create obstacles during the model training stage.
2. Limited base model selection. While CogVideoX-5B-I2V is a strong image-to-video model, it generally generates low-fps videos with suboptimal temporal coherence and limited motion dynamics, which might hinder the versatility of the proposed dataset. Expanding the dataset using newer and stronger video generators [1,2] could significantly improve the dataset's practicality.

[1] Wan T, Wang A, Ai B, et al. Wan: Open and advanced large-scale video generative models[J]. arXiv preprint arXiv:2503.20314, 2025.
[2] Kong W, Tian Q, Zhang Z, et al. Hunyuanvideo: A systematic framework for large video generative models[J]. arXiv preprint arXiv:2412.03603, 2024.

**Strengths Contributions:**

This paper proposes a large scale instruction-following video editing dataset, systematically organized into 18 diverse subcategories, covering both local and global edits, which is valuable and important to the development of this field.

---

> ### Author Rebuttal · Authors · 2025-07-31
>
> Thank you for your valuable feedback and recognition. Our **Señorita-2M** dataset is a large-scale, instruction-based video editing dataset containing approximately 2 million video triplets.
>
> This project took several months of dedicated effort, and we were committed to ensuring its quality and usefulness. To maintain high quality and strong text-video alignment, we collaborated with four professional video editors and leveraged high-perform open-source models. Complex user prompts were simplified into clear instructions using a large language model, and a robust filtering pipeline was applied to remove low-quality samples.
>
> Our dataset offers several key advantages:
>
> -   **First, it is both large and diverse**, covering two main categories and 18 sub-categories of common video editing tasks. It includes videos ranging from 33 to 64 frames, with resolutions between 336×592 and 1120×1984.
>
> -   **Second, it generalizes well across various base models**. We trained *CogVideo-5B*, *CogVideo-5B-I2V*, *Wan2.1-1.3B*, and *Wan2.1-14B-T2V*, all of which achieved strong editing performance using our dataset.
>
> -   **Third, it has been wellreceived by the community**, with over **15K downloads** on Hugging Face, including **4.5K in the last month**. Several recent papers have used Señorita-2M  training editors or developing unified generation and editing models, demonstrating its effectiveness and research value.
>
> &nbsp;
>
> ---
>
>
> ***According to the NeurIPS 2025 DB track policy, we are not permitted to include external links or updated PDFs during either the rebuttal or discussion stages. As a result, we are unable to provide additional visualization results. We sincerely appreciate your understanding.***
>
> ---
>
> &nbsp;
>
> **Q1： Imbalanced subcategories in the dataset. As illustrated in Table 2, "Object Addition" and "Object Removal" occupy 26.6% and 20.7% respectively, while some categories like "Outpainting" only constitute a small proportion at 2.0%. Such data distribution imbalance might introduce bias and create obstacles during the model training stage.**
>
>
> R1: Thank you for pointing out this issue. We'd like to clarify the following:
>
> *   **Some editing tasks (e.g., outpainting, deblur) are relatively easy to learn.** Tasks such as video outpainting and deblur are significantly easier than more complex tasks like object addition or removal. This is because the edited regions in these tasks are explicitly defined (e.g., a black masked area or the entire video), making them easier for the model to learn even from a smaller proportion of training examples.
>
> *   **Tasks with higher proportions (e.g., object removal, object addition) are more common in practice.** Since users are more likely to perform tasks such as object removal or stylization, we have placed greater emphasis on these during training.
>
> *   **The model performs well on the easier tasks without noticeable degradation.** Based on our evaluations, the model handles these simpler tasks effectively despite their lower percentage in the training set. *However, due to NeurIPS DB track policies, we are unable to provide visualizations.* Notably, previous image editing works such as Omni-Edit [1] and UltraEdit [2] also adopt imbalanced task distributions and report no significant performance drops across tasks. Thus, the use of an imbalanced task distribution does not appear to negatively affect overall model performance.
>
> [1]. OmniEdit: Building Image Editing Generalist Models Through Specialist Supervision. In ICLR 2025.
>
> Cong Wei, Zheyang Xiong, Weiming Ren, Xinrun Du, Ge Zhang, Wenhu Chen
>
> [2]. UltraEdit: Instruction-based Fine-Grained Image Editing at Scale. In NeurIPS 2024.
>
> Haozhe Zhao, Xiaojian Ma, Liang Chen, Shuzheng Si, Rujie Wu, Kaikai An, Peiyu Yu, Minjia Zhang, Qing Li, Baobao Chang
>
> &nbsp;
>
> **Q2： Limited base model selection. While CogVideoX-5B-I2V is a strong image-to-video model, it generally generates low-fps videos with suboptimal temporal coherence and limited motion dynamics, which might hinder the versatility of the proposed dataset. Expanding the dataset using newer and stronger video generators could significantly improve the dataset's practicality.**
>
> R2: Thank you for your suggestions. We would like to clarify two points:
>
> *  Our dataset construction pipeline can be applied with future base models. Besides of the dataset, our another contribution is the construction pipeline of the editing experts. Our pipeline is compatible with current base models, such as HunyuanVideo, CogVideoX and Wan2.1. Our pipeline is thoroughly designed and analyzed, enabling other researchers to follow our methodology and construct even stronger datasets with newer base models and larger video collections.
>
>
> * Our dataset is diverse and contains dynamic video content, with certain videos employing frame sampling at 2× or 4× rates to avoid the slow motion. Edited videos produced by our experts effectively retain these dynamic characteristics, enhancing temporal coherence and visual realism. This strategy also addresses the inherent low-fps limitations of some base models, ensuring the dataset robustly supports a broad spectrum of motion patterns.
>
>
> &nbsp;
>
> ---
>
> ***Thank you for your valuable comments to our paper. Your constructive suggestions are greatly appreciated.***

---

> > ### Comment · Reviewer_pNU1 · 2025-08-05
> >
> > Thanks for the reply. My concerns have been well addressed, so I prefer to maintain my positive score of accepting this paper.

---

> > > ### Author Response · Authors · 2025-08-05
> > >
> > > We sincerely appreciate your positive assessment and valuable feedback! Your comments have greatly encouraged us.

---

### Official Review · Reviewer_UHVn · 2025-07-01

**Rating:** 5
**Confidence:** 4

**Summary:**

This paper introduces Señorita-2M, a large-scale, high-quality, instruction-based video editing dataset comprising 2 million triplets of [source video, edited video, instruction]. To address the limitations of existing datasets in terms of edit diversity, frame consistency, and instruction alignment, the authors propose a systematic taxonomy of editing tasks (18 subcategories across local and global edits), curated using a mix of trained and existing task-specific experts. The dataset is further refined through a three-stage filtering pipeline involving quality classification, instruction alignment scoring, and visual similarity filtering. The paper also presents empirical validation through training multiple instruction-based editors on Señorita-2M, achieving state-of-the-art performance in temporal consistency, visual fidelity, and instruction alignment across multiple architectures and base models.

**Dataset Code Accessibility:**

Yes

**Dataset Code Comments:**

The dataset is accessible via Hugging Face.

**Ethical Considerations:**

No, there are no or only very minor ethics concerns

**Final Justification:**

The rebuttal has addressed my concerns. I keep my positive rating.

**Limitations Weaknesses:**

1. In general, video restoration is defined to enhance the low-quality video instead of filling in the black pixels. In the context of this work (Line 150), restoration refers to the process of inpainting. I suggest that authors correct "Restoration" with "Inpainting" to avoid ambiguity.
2. What are the common failure modes observed during inference with your trained models? Are there specific tasks or scenarios where performance drops noticeably?
3. How diverse and semantically rich are the LLM-generated instructions? Have you performed any linguistic or human annotation analysis to validate their naturalness and correctness?

**Strengths Contributions:**

1. In some tasks. e.g., Object swap, the edited video is treated as the source video while the original video is the target video. This is a clever idea to preserve the highest possible video quality.
2. The proposed dataset has the potential to advance the field of instruction-based video editing.

---

> ### Author Rebuttal · Authors · 2025-07-31
>
> Thanks for your valuable comments and recognition. Our Señorita-2M dataset is a large-scale instruction-based video editing dataset, consisting of around 2 millions of video triplets.
>
> The dataset is the result of a large and meaningful research project that took us several months to complete. We dedicated significant effort and focus to its development. To ensure high-quality video editing data and strong text-video alignment, we collaborated with four video editing experts. For generating simplified instructions, we used a LLM to extract them from complex user prompts. Additionally, we designed a filtering pipeline to remove low-quality samples and maintain the overall quality of the dataset.
>
> Generally speaking, our dataset has the following strengths. First, it is a large-scale and diverse dataset. It contains around 2 million video triplets from 2 main categories, with 18 sub-categories. It contains the most popular video editing tasks in our dataset. The resolution and frame length of the videos is diverse. Our dataset has videos from 33 to 64 frames, with the resolution of 336×592 to 1120×1984. Second, it has good generalization on different base models. We have trained CogVideo-5B, CogVideo-5B-I2V, Wan2.1-1.3B and Wan2.1-14B-T2V, respectively, and found our datasets can make good editors based on these pretrained models. Third, it has been popular in the community. We have recieved a lot of feedback from the community users. Currently, the dataset has been downloaded from the huggingface with *15K* times, with *4.5K* downloads in last month. Some papers used our dataset to build unified generation and editing model, they demonstrate that our dataset has a good effect for their model's construction.
>
> &nbsp;
>
> ---
>
> ***According to NeurIPS25 DB track, we are not allowed to put external links and updated pdfs in both rebuttal and discussion stage. Therefore, we couldn't put more visualization results. We sincerely hope your understanding.***
>
> ---
>
> **Q1: In general, video restoration is defined to enhance the low-quality video instead of filling in the black pixels. In the context of this work (Line 150), restoration refers to the process of inpainting. I suggest that authors correct "Restoration" with "Inpainting" to avoid ambiguity.**
>
> R1: Thanks for your advice. We will replace the restoration as the Inpainting in the revised paper. Your constructive suggestion help us produce better manuscripts.
>
> &nbsp;
>
> **Q2: The common failure modes and specific tasks or scenarios where performance drops noticeably.**
>
> **Q2.1: What are the common failure modes observed during inference with your trained models?**
>
> R2.1: The model trained on our dataset performs well on multiple tasks. However, we still notice it has limitations. *It has higher probability to fail on processing the complex actions. For example, a man is dancing on the ground, with complex action and fast movement. When attempting to replace the man with a tiger, the model frequently fails to generate a tiger that replicates the man's dance movements.*
>
> **Q2.2: Are there specific tasks or scenarios where performance drops noticeably?**
>
> R2.2: We find our model is struggling with *Counter-intuitive editing instructions*. For example, the instruction "Replace a person with a horse", often lead to failures. Since human walk with two legs while horses walk with four, the model becomes confused about how to position the horse's remaining two legs. We observe that these challenging edits are strongly correlated with the generative capabilities of the base model. Specifically, we find that even asking the base model to generate a horse walking on two legs is already very difficult. Therefore, we believe that  the ability of editor will increase with the developement of the generative models.
>
> &nbsp;
>
> **Q3: The diversity and semantically rich with the LLM-generated instructions. The linguistic or human annotation analysis of naturalness and correctness.**
>
> **Q3.1: How diverse and semantically rich are the LLM-generated instructions?**
>
> R3.1: Instead of only using the instruction that directly generated by LLM, we also ask LLM to generate the enhanced version. *For example, the remove tasks, we can find there are a lot of instructions in our dataset. *For example, "Delete {object}.", "Remove {object}.", "Discard {object}.", "Erase {object}.", "Clear {object}.", "Make {object} invisible.", "Remove {object} from the video."， "Make {object} disappear.", "Wipe out {object} from the video."* Therefore, by using this strategy, we make sure the diverse and semantically rich of the instructions that we provided in the dataset, preventing the overfitting risk of the editing model.
>
>
> **Q3.2: Have you performed any linguistic or human annotation analysis to validate their naturalness and correctness?**
>
> R3.2: Thanks for pointing out this question. Here, we want to clarify that:
>
> * When building the dataset, we have checked the llm-generated instructions with 1000 random sampled examples. Fortunately, all examples have precise instructions.
>
> * Generating precise instructions by LLM is reasonable, since the prompt reformulation is a relatively easy task for LLM to perform.  According to current LLM benchmark, we can find that most LLM have strong ability in language translation, paper rewriting and linguistic restatement [1, 2].
>
>
> [1] Measuring Massive Multitask Language Understanding. In ICLR 2021.
>
> Dan Hendrycks, Collin Burns, Steven Basart, Andy Zou, Mantas Mazeika, Dawn Song, Jacob Steinhardt
>
> [2] Beyond the imitation game: Quantifying and extrapolating the capabilities of language models. In Transactions on machine learning research.
>
> Srivastava, Aarohi and Rastogi, Abhinav and Rao, Abhishek and Shoeb, Abu Awal and Abid, Abubakar and Fisch, Adam and Brown, Adam R and Santoro, Adam and Gupta, Aditya and Garriga-Alonso, Adri and others
>
> &nbsp;
>
> ---
>
> **We sincerely thank you again for your valuable suggestions and hope our responses address your concerns clearly.**

---

> > ### Comment · Reviewer_UHVn · 2025-08-03
> > **Response to the authors' rebuttal**
> >
> > I have read the author's response. The author has addressed all my concerns, and the given failure example in Q2 is intuitive. I thus keep my positive rating.

---

> > > ### Author Response · Authors · 2025-08-03
> > >
> > > Thank you for your positive evaluation and constructive feedback. We are encouraged by your comments!

---

### Official Review · Reviewer_JiPv · 2025-07-08

**Rating:** 5
**Confidence:** 4

**Summary:**

This is a large scale instruction based dataset for video editing, which consists of 2M video instruction triplets. The dataset has 18 editing tasks and show effectiveness on multiple base model, e.g., CogVideoX5B and Wan2.1-1.3B. The authors have conducted sufficient experiments and provided editing samples in supplymentary files.

**Dataset Code Accessibility:**

Yes

**Ethical Considerations:**

No, there are no or only very minor ethics concerns

**Final Justification:**

Thank you for your reply, which satisfactorily addresses many of my earlier concerns. I am therefore inclined to accept the manuscript. Good luck.

**Limitations Weaknesses:**

- All source videos are scraped from Pexels, which may underrepresent complex real-world scenarios
-  Relying on task-specific specialists risks brittleness when new editing types arise; the paper does not evaluate how easily additional tasks can be integrated or how specialists trained on one domain generalize to unseen content.

**Strengths Contributions:**

- It consists of 2m video text triplets and has strong generalization on multiple base models.

- It successfully produced video editors and make nice editing videos. Experments show that it has effectiveness and could help researchers to build video editing models.

- The exploration of editing experts will help community to build video remover, inpainter and stylizer.

---

> ### Author Rebuttal · Authors · 2025-07-31
>
> Thanks for your valuable advice. Your comments help us to improve the quality of  our paper.
>
>
> Generally, we spent 9 months to build the dataset. We studied how to build the each video editing experts with DiT. *For example, we use the contrastive instructions to remove objects from the video. We use the image and video controlnet to jointly perform stylization, transferring the style information from the image editor to supply the lack of the video editor.* We use LLM to generate the instructions, and use our cleaning pipeline to filter out the useless video triplets. After these efforts, We successfully build a video dataset that is able to be used for editor training.
>
>
> Our dataset has the following advantages.
>
> * **The Señorita-2M is large-scale and diverse.** It consists of around 2 millions videos. The dataset can be broadly summarized as two categories, the global editing and local editing. The two categories can be split into 18 sub-categories.
>
> * **Our dataset generalizes well.** We trained our editing model with four base models, CogVideoX-5B, CogVideoX-5B-I2V, Wan2.1-1.3B and Wan2.1-14B-T2V. The experiments show that our dataset has good generalization across different base models. Moreover, the editing models can perform hard editing tasks, such as shape or size editing, which is the task that unseen in our training dataset.
>
> * **Our dataset is popular in the community.** We submitted our dataset on huggingface, and recieves **15K** downloads in total, **4.5K** downloads in last month. A lot of users also discussed with us via email. Some even use our model to train unified video generation and editing model, showing good editing results with their evaluation.
>
> &nbsp;
>
> ---
>
> ***According to NeurIPS25 DB track, we are not allowed to put external links and updated pdfs in both rebuttal and discussion stage. Therefore, we couldn't put more visualization results. We sincerely hope your understanding.***
>
> ---
>
> **Q1: All source videos are scraped from Pexels, which may underrepresent complex real-world scenarios.**
>
> R1: Thanks for your advice.
>
> The videos from Pexels have several advantages:
> * We use the videos that crawled from the Pexels which is a website that store images and videos which has good aesthetic effect. It can be shown that the model trained with data from the Pexels can have good aesthetic [1,2].
> * Using the videos in Pexels don't have legal risk. The Pexels has claimed that they allow users to download the videos and reproduce these videos are legal. Therefore, the reason we choose the videos from this website is that we can make our dataset legally, without the infringement concern.
>
> The model trained on our editing dataset performs well on most scenarios. We agree that our model may fail on some extreme scenarios. But using our dataset construction pipeline, researchers can build a dataset that with more complex real world scenarios.
>
>
> [1]. LAION‑Aesthetics. In LAION-website.
>
> [2]. FloVD: Optical Flow Meets Video Diffusion Model for Enhanced Camera-Controllable Generation. In CVPR 2025.
>
> Wonjoon Jin, Qi Dai, Chong Luo, Seung-Hwan Baek, Sunghyun Cho
>
> &nbsp;
>
> **Q2: Relying on task-specific specialists risks brittleness when new editing types arise; the paper does not evaluate how easily additional tasks can be integrated or how specialists trained on one domain generalize to unseen content.**
>
> R2: Thank you for pointing out this problem.
>
> **Q2.1: Relying on task-specific specialists risks brittleness when new editing types arise.**
>
> R2.1: The models trained on our dataset has good generalization ability on unseen tasks. We have evaluated our model on new editing tasks, e.g., shape editing, size editing, motion editing, multi-region editing, small object editing. Our model can generalize well on these tasks. These videos can be found in our supplementary file. You can unzip the supplementary file and click the *index.html* to view the demo page.
>
> **Q2.2: The paper does not evaluate how easily additional tasks can be integrated.**
>
> R2.2: During the early construction of the our dataset, the first version of dataset only contains the 5 most popular editing masks, we find that the editing model can only perform decent editing results. With more tasks were added in the dataset from 5 to 18 tasks, we found the editing ability of the model significantly increase. These 18 tasks cover the most common editing tasks in the real world.
>
>
> **Q2.3:  how specialists trained on one domain generalize to unseen content**
>
> R2.3: We have provided the visualization results on some complex tasks that our dataset does not involve. *For example, we have evaluated our model on new editing tasks, e.g., shape editing, size editing, motion editing, multi-region editing, small object editing.* Our model can generalize well on these tasks. These videos can be found in our supplementary file. You can unzip the supplementary file and click the *index.html* to view the demo page.
>
> &nbsp;
>
> ---
>
> **We sincerely thanks for your comments and advice. We hope our response can address your concern.**

---

> ### Author Response · Authors · 2025-08-08
>
> Dear Reviewer JiPv,
>
> Thank you for the valuable time and the effort in reviewing our paper.
>
> We hope you might find our responses satisfactory. We sincerely hope you will reconsider your rating based on our clarification in responses. Thank you for your time!
>
> Sincerely,
>
> Authors of submission 403

---

### Note · Authors · 2025-08-14

We sincerely thank all reviewers for their time and effort. Their constructive comments and thoughtful feedback improved our manuscript. Moreover, we deeply appreciate their recognition of our work, which has greatly encouraged us.

Our dataset offers three main advantages:

* **Large-scale and diverse.** The Señorita-2M dataset contains around 2 million video triplets, organized into global and local editing categories, further divided into 18 sub-categories.

* **Strong generalization.** Editing models trained on our dataset, across four base models (CogVideoX-5B, CogVideoX-5B-I2V, Wan2.1-1.3B, and Wan2.1-14B-T2V), demonstrated robust generalization. They can even perform difficult tasks, such as shape and size editing, which were unseen in training.

* **Community impact.** The dataset released on huggingface has received 15K total downloads, including 4.5K in last month (as of June 30, 2025). Many users have reached out with feedback, and some have successfully applied it to unified video generation and editing, showing promising results.

We hope Señorita-2M can serve as a valuable dataset for advancing research in video editing. Once again, we sincerely thank the reviewers for their insightful feedback and kind recognition, which not only improved this work but also motivate us to continue contributing to AIGC research.

---

### Decision · Program_Chairs · 2025-09-18

**Decision:**

Accept (poster)

**Comment:**

This paper presents Señorita-2M, a large-scale (2M triplets) instruction-based dataset for video editing, systematically organized into 18 subcategories with strong quality control. Models trained on it show superior fidelity, temporal consistency, and robust instruction alignment across multiple base models.

Strengths include the unprecedented scale, systematic taxonomy, reproducibility, and demonstrated community adoption. Weaknesses are dataset imbalance, reliance on Pexels videos, and occasional artifacts. However, these issues were acknowledged and reasonably addressed during rebuttal, with proposed improvements.

Given the novelty, rigor, and clear community impact, the work stands out within the DB track and merits a Spotlight.

===== FINAL UPDATE FROM DB Track PCs ====

The final decision for this paper has been taken by the program chairs after consultation with the SACs. All Senior Area Chairs have ranked papers according to the feedback from the AC during the review process. We decided to leave the original meta-review to reflect the opinion of the AC in light of the initial discussions with reviewers and SAC.